# MeSH: Memory-as-State-Highways for Recursive Transformers

**Chengting Yu**[†ζα], **Xiaobo Shu**[†α], **Yadao Wang**[α], **Yizhen Zhang**[α], **Haoyi Wu**[§α], **Jiaang Li**[α],
**Rujiao Long**[α], **Ziheng Chen**[α], **Yuchi Xu**[α], **Wenbo Su**[α], **Bo Zheng**[✉α]
[α] Alibaba Group    [ζ] Zhejiang University    [§] ShanghaiTech University

## Abstract

Recursive transformers reuse parameters and iterate over hidden states multiple times, decoupling compute depth from parameter depth. However, under matched compute, recursive models with fewer parameters often lag behind non-recursive counterparts. By probing hidden states, we trace this performance gap to two primary bottlenecks: **undifferentiated computation**, where the core is forced to adopt a similar computational pattern at every iteration, and **information overload**, where long-lived and transient information must coexist in a single hidden state. To address the issues, we introduce a *Memory-as-State-Highways (MeSH)* scheme, which externalizes state management into an explicit memory buffer and employs lightweight routers to dynamically diversify computation across iterations. Probing visualizations confirm that MeSH successfully resolves the pathologies by inducing functional specialization across iterations. On the Pythia suite (160M-6.9B), MeSH-enhanced recursive transformers consistently improve over recursive baselines and outperforms its larger non-recursive counterpart at the 1.4B scale, improving average downstream accuracy by +1.06% with 33% fewer non-embedding parameters. Our analysis establishes MeSH as a scalable and principled architecture for building stronger recursive models. Our code is available at *https://github.com/LivingFutureLab/MeSH/*.

## 1 Introduction

Scaling up model parameters and data has been a primary driver of improvements in the general capabilities of large language models (LLMs) (Kaplan et al., 2020; Hoffmann et al., 2022; Brown et al., 2020; Wei et al., 2022; Chowdhery et al., 2023; Grattafiori et al., 2024; OpenAI, 2023; Snell et al., 2024; Liu et al., 2024; Comanici et al., 2025) . However, further gains along this axis face headwinds: the supply of high-quality text is nearing exhaustion (Villalobos et al., 2022; Muennighoff et al., 2023) , empirical scaling curves show signs of saturation (Hackenburg et al., 2025; Hoffmann et al., 2022), and distributed pre-training incurs substantial, often super-linear, communication overheads as models grow (Narayanan et al., 2021; Pati et al., 2023; Li et al., 2024; Patterson et al., 2021; Momeni et al., 2025).

As a parameter-efficient architectural response to the scaling bottlenecks of large models, recursive transformers have recently attracted growing interest (Geiping et al., 2025; Bae et al., 2024; 2025; Zeng et al., 2025; Saunshi et al., 2025). The core idea behind is to decouple computational depth from parameter depth by repeatedly applying a compact, weight-shared core block in a loop. By breaking the tight coupling between these two depths, recursive transformers natively enable dynamic computation: they can, in principle, allocate computational budgets adaptively based on task difficulty to reduce inductive bias (Bae et al., 2025), and open up a new scaling axis of computational depth, complementing model size and data volume (Zhu et al., 2025b; Geiping et al., 2025; Saunshi et al., 2025).

However, a critical challenge remains: under matched compute, recursive models with fewer parameters often lag behind their non-recursive counterparts (i.e., they exhibit higher perplexity or lower accuracy compared to standard Transformers with equivalent FLOPs but unique parameters

---

[†] Equal contribution    [✉] Corresponding author

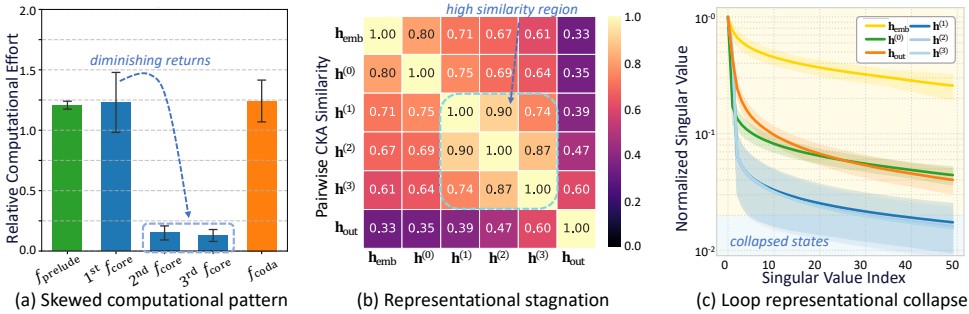

Figure 1: **Diagnostic visualizations of the recursive transformer.** Analyses are performed on a Pythia-410M-based model with the Prelude-Reccurent-Coda architecture (3 core loops). Hidden state matrices ($\mathbf{h} \in \mathbb{R}^{\text{seq} \times \text{dim}}$) are extracted from 500 samples from the Pile dataset. The states $\mathbf{h}_{\text{emb}}, \mathbf{h}^{(0)} \dots \mathbf{h}_{\text{out}}$ refer to the initial token embeddings, the states to each block, and the final output state. We leave further experimental details and analysis to Section 4.1. **(a) Skewed computational pattern.** Plots the relative magnitude of the state update, calculated for each computational block ($f$) as $2||f(\mathbf{h}) - \mathbf{h}||_F / (||f(\mathbf{h})||_F + ||\mathbf{h}||_F)$, where $||\cdot||_F$ is the Frobenius norm, which serves as a proxy for the computational effort of each block. The x-axis tracks the flow from **Prelude** through consecutive **Core** iterations to **Coda** within the common *Prelude-Recurrent-Coda* structure (Geiping et al., 2025). Bars show the mean and standard deviation across 500 samples. **(b) Representational stagnation.** Displays the pairwise Centered Kernel Alignment (CKA) (Kornblith et al., 2019) similarity with an RBF kernel between the hidden state matrices. High CKA values between consecutive loop states ($h^{(t)}, h^{(t+1)}$) indicate that the representation has ceased to evolve, trapping the model in a fixed point. **(c) Loop representational collapse.** Shows the top 50 normalized singular values ($\sigma_i/\sigma_0$) for each hidden state matrix on a logarithmic Y-axis. The decay rate of the spectrum indicates the effective rank or intrinsic dimensionality of each state matrix.

per layer). In this work, we provide measurable evidence that the performance gap stems from fundamental bottlenecks: **undifferentiated computation** and **information overload**, quantified by three observables as skewed computation, representational stagnation, and dimensional collapse. To address the pathologies, we propose the *Memory-as-State-Highways (MeSH)* scheme, a principled architectural modification that replaces the overloaded hidden state with an explicit memory buffer governed by lightweight, step-wise routers. The proposed design separates persistent memory from transient computation, effectively converting the implicit challenge of state management into a clear, learnable routing problem for recursive transformers.

## 2 WHY NAIVE RECURSION FAILS: A DIAGNOSTIC ANALYSIS

The core premise of a recursive transformer is to reuse a weight-shared computational block, yet the design introduces a fundamental limitation: the block lacks any explicit information about its progress within the iterative sequence, which prevents effective functional specialization and leads to inefficient computation. This also forces a single hidden state to handle multiple conflicting information. We define these two primary bottlenecks as **undifferentiated computation** and **information overload**.

**Undifferentiated computation.** The inability to differentiate between loop steps prevents the model from assigning specialized roles to each iteration. This leads to two distinct failure modes. First, the model exhibits a *skewed computational pattern*, as shown in Figure 1a. The first core loop performs the vast majority of the computational work, while the update magnitudes of subsequent iterations drop to near zero. This sharp decay suggests that the features stabilize prematurely, indicating that the model fails to effectively utilize the computational depth of later loops or distribute its processing logic over multiple steps. Second, consecutive loop states exhibit high representational similarity, indicating *representational stagnation* (Figure 1b). We employ Centered Kernel Alignment (CKA) (Kornblith et al., 2019) to measure this, as it provides a robust similarity metric invariant to orthogonal transformations. High CKA similarity (Kornblith et al., 2019) between con-

secutive loop states (i.e., $\mathbf{h}^{(t)} \approx \mathbf{h}^{(t+1)}$) reveals that the model becomes trapped in a fixed-point attractor, repeatedly applying a near-identical transformation instead of progressively refining its representation.

**Information overload.** In principle, coordinating multiple functional roles across recursive steps benefits from higher-dimensional representations; under naïve recursion, however, the model is constrained to a single hidden state, making such separation difficult in practice. Concurrently, the single hidden state vector is forced to be the sole carrier for all information, creating a severe bottleneck, where the single state could be forced to manage multiple, often conflicting, roles simultaneously:

- **Long-term Memory:** Preserving key information from the initial input to prevent catastrophic forgetting and maintaining stability across repeated iterations.
- **Working Memory:** Preparing intermediate features for the subsequent iteration and supporting high-plasticity, transient computations within each step.

This single-state constraint induces a trade-off between stability and plasticity. Empirically, naïve recursive models tend to prioritize long-term stability to avoid forgetting, leading the representation to converge toward a stable, shared "common ground" that favors persistence over flexible processing. The information overload on the hidden state forces the model to find a low-dimensional "common ground" representation that can safely survive multiple transformations, which directly causes *loop representational collapse*. We quantify this by analyzing the normalized singular value spectrum of the hidden state matrices, a proxy for their effective rank (see Figure 1c). The singular value spectrum of the loop states decays much more rapidly than that of the initial state, indicating a collapse into a lower-dimensional subspace and a significant loss of expressive capacity. Complementing this, a subspace analysis across iterations shows that the variance of hidden states concentrates within a shared, low-dimensional subspace—consistent with a dominant long-term component (Appendix E.8). Taken together, these observations are consistent with the hypothesis that, under naïve recursion, information overload biases the system toward a stable, low-rank manifold that preserves global context at the expense of the high-dimensional capacity required for transient, stepwise processing.

The diagnosis of undifferentiated computation and information overload directly motivates our solution, MeSH, which is specifically designed to address these identified problems.

## 3 METHODOLOGY: ALLEVIATING INFORMATION OVERLOAD AND ENABLING FUNCTIONAL SPECIALIZATION

The pathologies diagnosed in Section 2—undifferentiated computation and information overload—stem from the architectural limitations of naive recursion. In this section, we develop a methodology aimed at alleviating these core issues. We first review common heuristic-based recurrence schemes, which use fixed, additive connections to supplement the context at each step, that can be seen as attempts to mitigate **information overload** but do little to address the problem of **undifferentiated computation**. We then introduce our proposed solution, MeSH, a general framework designed to systemically alleviate both information overload and the lack of functional specialization.

### 3.1 PRELIMINARIES: ARCHITECTURE OF RECURSIVE TRANSFORMERS

**Overall Architectural Structure.** Recursive transformers achieve computational depth by repeatedly applying a shared, weight-tied **core** block, $f_{\text{core}}(\cdot)$. The central idea is to refine a hidden state $\mathbf{h}^{(t)} \in \mathbb{R}^{L \times D}$ (where $L$ is the sequence length and $D$ is the hidden dimension) over a sequence of $K$ iterations. Starting from an initial state $\mathbf{h}^{(0)}$, the simplest form of recurrence updates the state as $\mathbf{h}^{(t+1)} = f_{\text{core}}(\mathbf{h}^{(t)})$. The core recurrence loop could be embedded within a broader network topology that defines how the initial state $\mathbf{h}^{(0)}$ is produced and how the final state $\mathbf{h}^{(K)}$ is consumed. We adopt the ***Prelude-Recurrent-Coda*** structure (Geiping et al., 2025) (also called ***Middle-Cycle*** (Bae et al., 2025)), which augments the core recursive block with specialized, non-tied prelude and coda networks. The framework first uses a **prelude** block, $f_{\text{pre}}$, to process the initial token embeddings ($\mathbf{h}_{\text{emb}}$) and prepare the first state for the loop: $\mathbf{h}^{(0)} = f_{\text{pre}}(\mathbf{h}_{\text{emb}})$. The recursive loop then runs for $K$

steps, after which its final output state, $\mathbf{h}^{(K)}$, is passed to a **coda** block, $f_{\text{coda}}$, to produce the model's final representation: $\mathbf{h}_{\text{final}} = f_{\text{coda}}(\mathbf{h}^{(K)})$.

**Core Recurrence Variants.** While the base recurrence, $\mathbf{h}^{(t+1)} = f_{\text{core}}(\mathbf{h}^{(t)})$, represents a straightforward cascade of computations, it may struggle with information retention, as each iteration can overwrite or forget crucial aspects of its input. To alleviate this representational burden on the core, the state update can be augmented with historical information. We summarize several common variants, which represent different strategies for information propagation (illustrated in Figure 2). The general update rule with such a context supplement is:

$$\mathbf{h}^{(t+1)} = f_{\text{core}}(\mathbf{h}^{(t)}) + \mathbf{h}_{\text{sup}}^{(t)} \tag{1}$$

where $\mathbf{h}_{\text{sup}}^{(t)}$ is a supplementary context. Common choices for this context include:

- **Residual:** Setting $\mathbf{h}_{\text{sup}}^{(t)} = \mathbf{h}^{(t)}$ introduces a standard skip connection between adjacent iterations, which allows the core to learn a residual update, enabling the model to incrementally refine the representation and aggregate information from all preceding steps (Yu et al., 2025; Zeng et al., 2025; Bae et al., 2025).

- **Anchor:** Setting $\mathbf{h}_{\text{sup}}^{(t)} = \mathbf{h}^{(0)}$ explicitly tethers each iteration to the initial state that entered the loop. The intuition is to prevent the iterative process from drifting too far from the initial semantics by continuously reinforcing the starting context (Yang et al., 2023; Mohtashami et al., 2023; Geiping et al., 2025).

- **Anchor\*:** An alternative, $\mathbf{h}_{\text{sup}}^{(t)} = \mathbf{h}_{\text{emb}}$, anchors each iteration to the raw token embeddings.

The heuristic connectivity schemes can be seen as attempts to mitigate **information overload**. By providing a direct, additive path for historical information (like the initial state $\mathbf{h}^{(0)}$ or the previous state $\mathbf{h}^{(t)}$), they partially offload the burden of memory from the main hidden state pathway. Note that these refer strictly to state-passing mechanisms to improve information flow, distinct from parameter-addition methods like that specialize weights (Bae et al., 2024). This allows the core to focus more on transformation rather than just preservation. However, it is important to note that these are rigid, non-adaptive solutions. The choice among them is often a **heuristic design decision** that requires careful empirical validation. Crucially, they do little to address the problem of **undifferentiated computation**, as the core block remains blind to its position in the loop.

### 3.2 MeSH: Memory–as-State-Highways for Recursive Transformers

We move beyond fixed recurrence rules by introducing the *Memory–as-State-Highways (MeSH)*, a mechanism that replaces the simple state-passing scheme. As shown in Figure 2, MeSH externalizes state management into an explicit state buffer controlled by learnable, step-wise read-write routers. This design decouples transient computation within the recursive core from persistent memory, allowing the model to dynamically manage iteration-specific information flow. The MeSH-augmented loop consists of several lightweight components:

**1. State Buffer and Initialization.** MeSH maintains a state buffer $\mathbf{M}$ with $B$ memory slots, $\mathbf{M} = \{\mathbf{m}_0, \mathbf{m}_1, \ldots, \mathbf{m}_{B-1}\}$, where each slot $\mathbf{m}_b \in \mathbb{R}^{L \times D}$ shares the same dimensions as the hidden states. Before the loop begins, the buffer is initialized by placing the raw token embeddings, $\mathbf{h}_{\text{emb}}$, into the first slot. This designated slot, $\mathbf{m}_0$, serves as a initial anchor to the original input. All other slots are initialized to zero:

$$\mathbf{m}_0^{(0)} = \mathbf{h}_{\text{emb}}, \quad \text{and} \quad \mathbf{m}_{b>0}^{(0)} = 0 \tag{2}$$

**2. Core Computation and Dynamic Routers.** The core block $f_{\text{core}}$ remains the central computational unit. The interface to the buffer is managed by step-wise **Write and Read Routers** ($R_{\text{write}}^{(t)}$ and $R_{\text{read}}^{(t)}$), which have unique parameters for each iteration $t = 0, \ldots, K - 1$. At each step, they compute routing weights based on the current hidden state $\mathbf{h}^{(t)} \in \mathbb{R}^{L \times D}$:

$$\mathbf{w}_{\text{write}}^{(t)} = \text{Softmax}(\text{Linear}_{\text{write}}^{(t)}(\mathbf{h}^{(t)})), \quad \mathbf{w}_{\text{read}}^{(t)} = \text{Softmax}(\text{Linear}_{\text{read}}^{(t)}(\mathbf{h}^{(t)})) \tag{3}$$

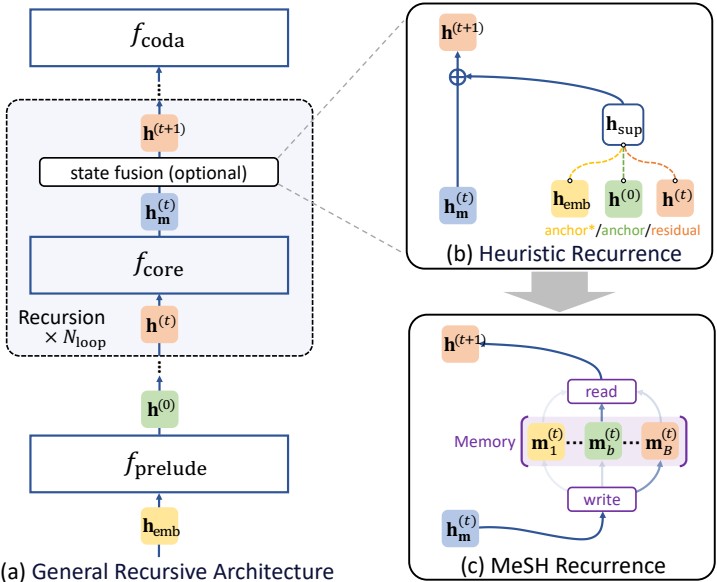

Figure 2: **Comparison of recurrence schemes. (a)** The general architecture of a recursive transformer involves the general dataflow passing a state $\mathbf{h}^{(t)}$ through a core computational block $f_{\text{core}}$ to produce the next state $\mathbf{h}^{(t+1)}$. **(b)** Common heuristic variants employ a fixed, additive state update to optimize the information flow, where the core output is supplemented by historical states $\mathbf{h}_{\text{sup}}$ (e.g., initial state $\mathbf{h}^{(0)}$ for `anchor` or previous state $\mathbf{h}^{(t)}$ for `residual`). **(c)** Our proposed MeSH replaces this rigid addition with a dynamic memory mechanism, which explicitly manages historical states via learnable write and read operations, allowing the model to flexibly retrieve and combine information to form the next state $\mathbf{h}^{(t+1)}$.

Each $\text{Linear}^{(t)}$ function is a one-layer projection that maps the $D$-dimensional hidden state of each token to a vector of $B$ logits, corresponding to the number of buffer slots. A softmax function is then applied along the slot dimension for each token to normalize these logits, producing the final weight matrices $\mathbf{w}_{\text{write}}^{(t)}$ and $\mathbf{w}_{\text{read}}^{(t)}$, both of shape $\mathbb{R}^{L \times B}$.

**3. MeSH-Augmented Recurrence and Integration.** The fixed context supplementation is replaced by a memory update logic, as illustrated in Figure 2b. At each step $t$, the core first computes its output $\mathbf{h}_{\text{m}}^{(t)}$ from the current state $\mathbf{h}^{(t)}$:

$$\mathbf{h}_{\text{m}}^{(t)} = f_{\text{core}}(\mathbf{h}^{(t)}) \tag{4}$$

The buffer is then updated via a distributed write operation for a soft insertion of the state, where the output $\mathbf{h}_{\text{m}}^{(t)}$ is scaled by the computed write weights before being added to the memory slot:

$$\mathbf{m}_b^{(t+1)} = \mathbf{m}_b^{(t)} + \mathbf{h}_{\text{m}}^{(t)} \odot \mathbf{w}_{\text{write},b}^{(t)}, \quad \text{for } b = 0, \ldots, B-1 \tag{5}$$

where $\odot$ denotes element-wise multiplication with broadcasting. Subsequently, the state for the next iteration, $\mathbf{h}^{(t+1)}$, is synthesized via a read operation from the updated buffer:

$$\mathbf{h}^{(t+1)} = \sum_{b=0}^{B-1} \mathbf{m}_b^{(t+1)} \odot \mathbf{w}_{\text{read},b}^{(t)}$$

In the **prelude-recurrent-coda** setting, a dedicated transitional write-read cycle first processes the prelude's output $f_{\text{pre}}(\mathbf{h}_{\text{emb}})$ to synthesize the initial state $\mathbf{h}^{(0)}$. After the main loop, a final read operation computes the output $\mathbf{h}^{(K)}$ from the memory buffer before passed to the coda. The full computational process is detailed in the pseudocode in Appendix C.

### 3.3 How MeSH Addresses the Diagnosed Pathologies

The architectural design of MeSH, centered on state externalization and dynamic routing, directly counteracts the core pathologies diagnosed in Section 2.

**Enabling Functional Specialization via Dynamic State Composition.** MeSH explicitly breaks the cycle of **undifferentiated computation** by replacing the rigid, additive update rule of heuristic methods with a dynamic read-write cycle controlled by step-wise routers. Since each router $(R_{\text{write}}^{(t)}, R_{\text{read}}^{(t)})$ has its own unique set of learnable parameters for each iteration $t$, the model is no longer forced to apply a single, universal transformation. Instead, at each step, it learns to dynamically synthesize the next state by retrieving a context-specific mixture of information from the memory buffer, which contains all relevant historical states. This flexibility allows MeSH to learn and dynamically switch between complex recurrence behaviors. The ability to adapt the recurrence rule on the fly is the implicit mechanism to enables functional specialization.

**Alleviating Information Overload via State Externalization.** MeSH directly alleviates **information overload** by decoupling persistent memory from transient computation. The external state buffer $\mathbf{M}$ serves as a dedicated, multi-slot highway for long-lived information. This design relieves the primary hidden state $\mathbf{h}^{(t)}$ from the burden of simultaneously storing historical context and serving as the workspace for the core block. The hidden state can now utilize its full dimensionality for complex, transient computations, knowing that essential long-term information is safely preserved in the buffer and can be retrieved on demand by the read router. This allows the model to maintain high-dimensional, expressive representations throughout the entire iterative process.

In essence, MeSH replaces the single, overloaded information channel of standard recurrence with a multi-slot memory buffer and dynamic, state-aware routers. The principled design provides a systemic and highly expressive solution to the core problems in recursive transformers, subsuming prior heuristic approaches into a more general framework (see Appendix D for more discussion).

## 4 Experiments

We pretrain our models from scratch, closely following the methodology of the Pythia suite (Biderman et al., 2023). We employ the same GPT-NeoX-based architecture and train on a deduplicated subset of the Pile dataset (Gao et al., 2020), curated by EleutherAI. For evaluation, we assess two primary aspects of model performance. We report perplexity scores on the validation sets of the Pile (Gao et al., 2020), Wikitext, and the Lambada (both OpenAI and Standard versions) datasets (Paperno et al., 2016) to measure language modeling capabilities. We also evaluate downstream performance on a suite of 9 few-shot benchmarks using the LM Evaluation Harness framework (Gao et al., 2024). Detailed training configurations and evaluation procedures are described in Appendix B.

### 4.1 A Comparative Diagnostic Analysis of Recurrence Schemes

In Section 2, we identified three critical symptoms arising from naive recursive transformers: a skewed computational pattern, representational stagnation, and loop representational collapse. We conduct a detailed analysis of the internal dynamics of four model variants: a `base` recursive model, two common heuristic variants (`+residual` and `+anchor`), and our proposed `+mesh` architecture. The analysis is performed on a Pythia-410M model with configuration of `3+6R3+3`, averaging results over 500 samples from the Pile dataset.

**MeSH mitigates the skewed computational pattern.** Figure 3 visualizes the computational effort of each block, confirming the pathology of naive recursion. The `base` model exhibits an extreme computational imbalance: the first core loop ($1^{\text{st}} f_{\text{core}}$) accounts for the vast majority of the work, while subsequent loops contribute negligibly, demonstrating a classic case of **diminishing returns**. While the `+residual` and `+anchor` heuristics offer partial relief, the computational effort still decays sharply. In stark contrast, the `+mesh` model achieves a remarkably **balanced computational distribution**, with all three core loops contributing significantly. This demonstrates that MeSH's

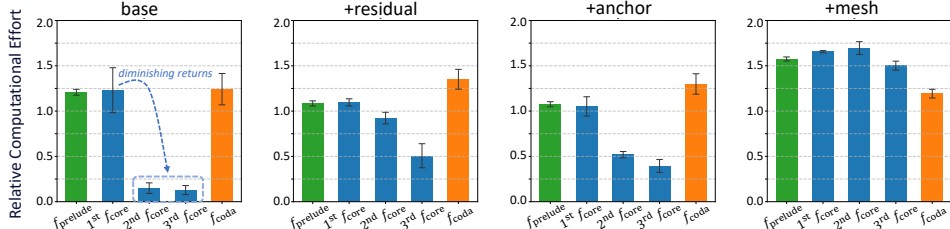

Figure 3: **Skewed Computational Pattern.** Plots the relative magnitude of the state update, calculated for each computational block ($f$) as $2||f(\mathbf{h}) - \mathbf{h}||_F/(||f(\mathbf{h})||_F + ||\mathbf{h}||_F)$, where $|| \cdot ||_F$ is the Frobenius norm, which serves as a proxy for the computational effort of each block. Bars show the mean and standard deviation across 500 samples.

dynamic read-write mechanism endows the model with a sense of iterative progress, allowing it to assign distinct and meaningful computational roles to each step.

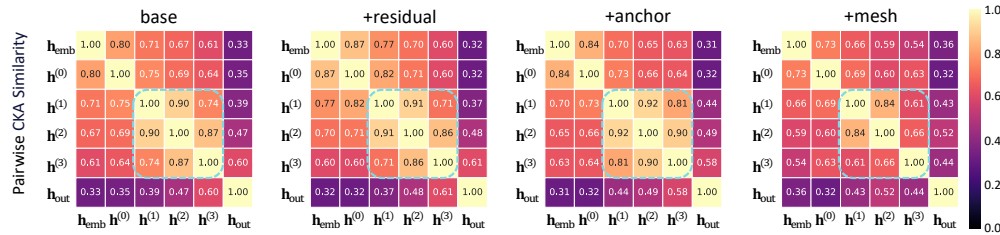

Figure 4: **Representational Stagnation.** Displays the pairwise Centered Kernel Alignment (CKA) similarity with an RBF kernel between hidden state matrices ($\mathbf{h} \in \mathbb{R}^{\text{seq} \times \text{dim}}$) at different stages of the model. The matrix shows the mean similarity across 500 samples. High similarity (values near 1.0) between consecutive loop states indicates that representations have stopped evolving.

**MeSH breaks representational stagnation.** Figure 4 displays the CKA similarity (Kornblith et al., 2019) between hidden states. High similarity between consecutive loop states ($h^{(1)}$, $h^{(2)}$, $h^{(3)}$) signals that the model is trapped in a fixed-point attractor. The `base` model's loop states exhibit very high CKA similarity, confirming severe **representational stagnation**. The `+mesh` model **reduces the similarity between consecutive loop states**, proving it has broken free from stagnation while its memory buffer allows it to maintain a strong connection to the initial context.

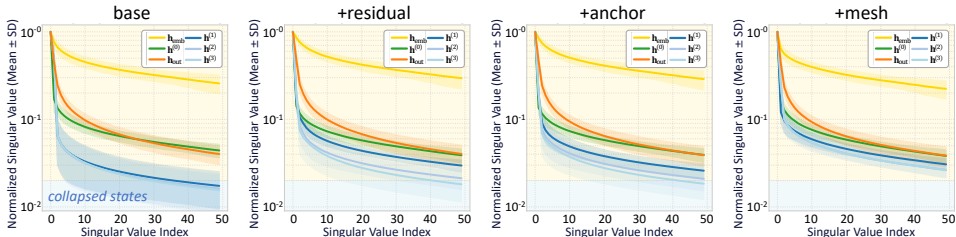

Figure 5: **Loop Representational Collapse.** Shows the top 50 normalized singular values ($\sigma_i/\sigma_0$) for key hidden state matrices on a logarithmic Y-axis. The decay rate of the spectrum indicates the effective rank or intrinsic dimensionality of each state matrix. A faster decay signifies a collapse into a lower-dimensional representation. Lines and shaded areas represent the mean and standard deviation across 500 samples.

**MeSH prevents loop representational collapse.** Figure 5 plots the singular value spectrum for hidden states. In the `base` model, the loop states ($h^{(1)}$, $h^{(2)}$, $h^{(3)}$) show a much faster spectral decay than the input state ($h^{(0)}$), confirming **loop representational collapse** into a low-dimensional subspace as a result of a forced "representational compromise". The heuristic fixes offer only marginal

Table 1: **Comparison of MeSH, Recursive and Vanilla Transformers.** Performance is measured by perplexity (PPL↓) on four datasets and average accuracy (Avg. acc↑) on a suite of 10 downstream tasks. The percentage reduction in non-embedding parameters for recursive models is shown in parentheses. The `Layers` for recursive models follow the format '$\{L_{\text{prelude}}\}+\{L_{\text{core}}\}R\{N_{\text{loop}}\}+\{L_{\text{coda}}\}$', indicating the number of layers in the prelude, core, coda. $\Delta$acc shows the absolute accuracy change relative to the `Vanilla` non-recursive baseline. `LD-O` and `LD-S` refer to Lambada OpenAI and Standard. The best results within each recursive block are **bolded**, and second best results are underlined.

| | Structure | | | Perplexity↓ | | | | Task Avg. acc↑ / $\Delta$acc (%) | |
| --- | --- | --- | --- | --- | --- | --- | --- | --- | --- |
| | Scheme | Layers | Variant | Pile | Wiki | LD-O | LD-S | 0-shot | 5-shot |
| **160M** | Vanilla | 12 | — | 11.31 | 30.32 | 42.86 | 129.89 | 39.88 | 40.54 |
| | Recursive (-33.3%) | 2+4R2+2 | base | 11.79 | 32.32 | 53.06 | 217.87 | 38.90 / -0.98 | 39.29 / -1.25 |
| | | | +anchor | 11.63 | 31.69 | 50.38 | 195.11 | 38.81 / -1.07 | 40.15 / -0.39 |
| | | | +mesh | **11.37** | **30.43** | **46.60** | **178.77** | **39.41** / -0.47 | **40.60** / +0.06 |
| **Pythia-410M** | Vanilla | 24 | — | 9.07 | 21.79 | 19.48 | 65.86 | 43.87 | 45.31 |
| | Recursive (-33.3%) | 4+8R2+4 | base | 9.31 | 22.74 | 22.57 | 53.76 | 43.26 / -0.61 | 45.03 / -0.28 |
| | | | +anchor | 9.19 | 22.12 | 20.37 | 52.55 | 43.70 / -0.17 | **45.68** / +0.37 |
| | | | +mesh | **9.09** | **21.84** | **19.63** | **42.51** | **44.12** / +0.25 | 45.56 / +0.25 |
| | Recursive (-50.0%) | 3+6R3+3 | base | 9.65 | 23.88 | 26.76 | 81.75 | 41.94 / -1.93 | 44.01 / -1.30 |
| | | | +residual | 9.69 | 24.05 | 26.31 | 76.76 | 42.16 / -1.71 | 44.24 / -1.07 |
| | | | +anchor | 9.49 | 23.31 | 24.49 | 72.30 | 42.85 / -1.02 | 44.90 / -0.41 |
| | | | +mesh | **9.35** | **22.80** | **20.72** | **52.07** | **43.53** / -0.34 | **46.04** / +0.73 |
| **Pythia-1B** | Vanilla | 16 | — | 7.96 | 17.66 | 13.53 | 33.65 | 46.95 | 49.07 |
| | Recursive (-31.3%) | 3+5R2+3 | base | 8.20 | 18.64 | 14.44 | 36.39 | 45.72 / -1.23 | 47.75 / -1.32 |
| | | | +residual | 8.19 | 18.46 | 14.18 | 35.54 | 46.19 / -0.76 | 47.85 / -1.22 |
| | | | +anchor* | 8.07 | 18.06 | 12.90 | 30.56 | 46.85 / -0.10 | 49.18 / +0.11 |
| | | | +anchor | 8.10 | 18.15 | 13.32 | 32.34 | 46.73 / -0.22 | 48.83 / -0.24 |
| | | | +mesh | **7.90** | **17.54** | **12.19** | **26.71** | **47.53** / +0.58 | **49.51** / +0.44 |
| **Pythia-1.4B** | Vanilla | 24 | — | 7.44 | 15.97 | 10.51 | 22.81 | 49.50 | 51.93 |
| | Recursive (-33.3%) | 4+8R2+4 | base | 7.63 | 16.64 | 11.38 | 23.69 | 48.89 / -0.61 | 50.99 / -0.94 |
| | | | +residual | 7.58 | 16.44 | 10.91 | 20.44 | 49.50 / +0.00 | 51.18 / -0.75 |
| | | | +anchor* | 7.51 | 16.27 | 10.81 | **19.14** | 49.29 / -0.21 | 51.83 / -0.10 |
| | | | +anchor | 7.51 | 16.25 | 10.71 | 19.37 | 49.39 / -0.11 | 51.27 / -0.66 |
| | | | +mesh | **7.39** | **15.84** | **9.72** | 19.39 | **50.56** / +1.06 | **52.79** / +0.86 |

gains. The `+mesh` model, however, demonstrates the ability to **preserve representational richness**, allowing the hidden state to maintain a high-dimensional, expressive structure throughout the iterative process.

## 4.2 Main Results

We conducted experiments on Pythia models ranging from 160M to 1.4B parameters, creating recursive variants with approximately 33% fewer non-embedding parameters to compare against standard `Vanilla` baselines (non-recursive models with unique parameters) and simpler recursive schemes (Table 1). While naive recursion (`base`) degrades performance and fixed schemes like `+anchor` offer only partial recovery, our MeSH-enhanced models (`+mesh`) consistently outperform all other variants. The `+mesh` models can even surpass their larger, more parameter-heavy `Vanilla` counterparts. For instance, the Pythia-1.4B `+mesh` model, despite its smaller footprint, improves 0-shot and 5-shot average accuracy by +1.06% and +0.86% respectively over the `Vanilla` version, while also achieving state-of-the-art perplexity scores across all datasets. Furthermore, the performance advantage scales favorably with model size, confirming that MeSH's dynamic state management is not only effective but also a highly efficient and scalable architectural principle.

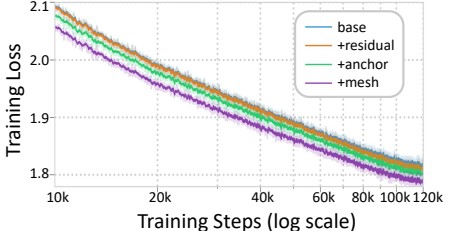 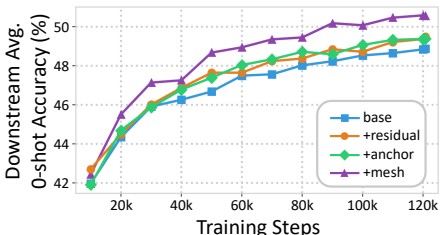

Figure 6: **Training Dynamics of Recursive Variants.** Comparison of training loss and downstream 0-shot accuracy for the 1.4B-Pythia-based recursive models. **(Left)** Training loss curve over 120k steps on a logarithmic x-axis. **(Right)** Downstream average 0-shot accuracy evaluated at checkpoints along a linear x-axis.

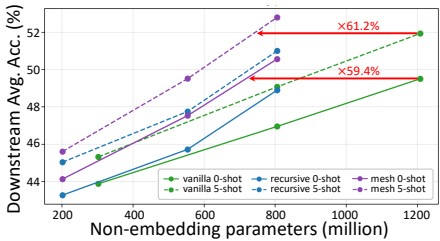 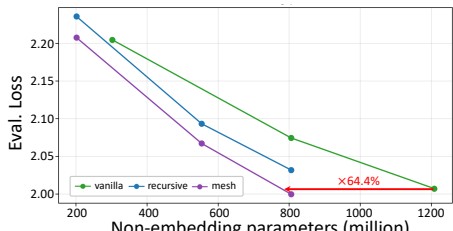

Figure 7: **Scaling Analysis of MeSH vs. Baselines.** Performance of `Vanilla` (non-recursive), naive `Recursive`, and `+mesh` models plotted against non-embedding parameter counts. **(Left)** Average downstream accuracy (0-shot and 5-shot). **(Right)** Evaluation loss.

## 4.3 FURTHER ANALYSIS AND ABLATION STUDIES

**Analysis of Training Dynamics**. To understand not just the final performance but also the learning process itself, we visualize the training dynamics of the 1.4B-parameter recursive variants in Figure 6. By juxtaposing pre-training loss with downstream accuracy evaluated at various checkpoints, we can assess both the learning efficiency and the rate at which models acquire useful capabilities. The training loss curves (Figure 6, left panel) reveal that the `+mesh` model consistently achieves a lower loss throughout pre-training. This indicates superior learning efficiency, as MeSH is able to fit the training data more effectively at every stage compared to the `base`, `+residual`, and `+anchor` variants. The training advantage translates directly into stronger downstream performance. The right panel of Figure 6 shows that the `+mesh` model not only starts from a stronger initial checkpoint but also exhibits a steeper and more consistent improvement in 0-shot accuracy. The superiority provides compelling evidence that MeSH's architectural modifications fundamentally enhance the model's ability to acquire and retain useful knowledge throughout the entire pre-training process, rather than being just a final-step improvement.

**Scaling Properties and Parameter Efficiency**. We provide scaling results in Figure 7, revealing the parameter efficiency of the MeSH architecture. While naive `recursive` models (blue lines) consistently underperform their standard `vanilla` counterparts (green lines) despite saving about 33% of parameters, our `+mesh` models (purple lines) not only decrease the performance degradation but could even outperform the `Vanilla` baselines at large scales. For example, our 805M-parameter `+mesh` model achieves 50.6% (0-shot) and 52.8% (5-shot) accuracy, surpassing the 1.2B-non-emb-parameter `Vanilla` model's 49.5% (0-shot) and 51.9% (5-shot), which translates to a **1.46x improvement in parameter efficiency**, allowing a MeSH-enhanced model to achieve the same level of evaluation loss as a `Vanilla` model with almost **a third fewer parameters**.

**Impact of Layer Distribution and Parameter Scaling.** To further dissect the architectural benefits of MeSH, we conduct a control study on the distribution of layers within the prelude-loop-coda framework. Using the Pythia-410M architecture as a testbed, we train several recursive models with varying configurations while keeping the total compute equivalent to the 24-layer non-recursive `Vanilla` model. We plot the validation perplexity against the percentage of non-embedding pa-

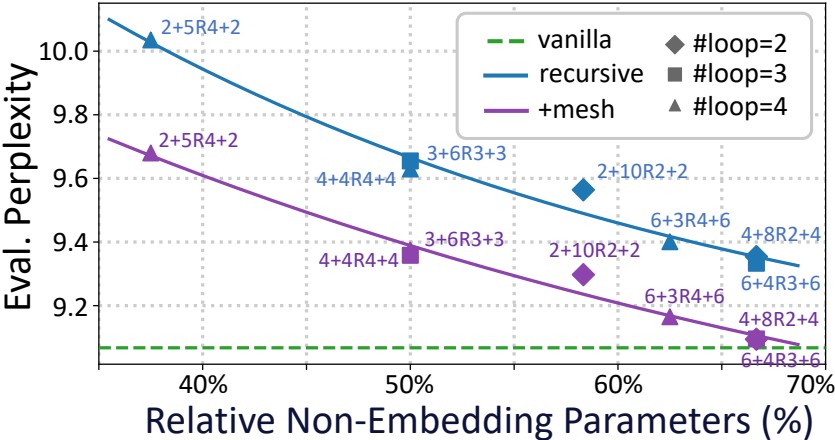

Figure 8: **PPL vs. Parameter Efficiency for Pythia-410M.** The plot shows the Pile perplexity as a function of non-embedding parameters, shown as a percentage relative to the `Vanilla` baseline. Each point represents a different distribution of layers (prelude, core, coda). The total computational depth for all models is aligned with the 24-layer non-recursive `Vanilla`.

rameters relative to the `Vanilla` baseline, with the results shown in Figure 8. The `+mesh` architecture (purple line) consistently achieves lower perplexity than the baseline `recursive` model (blue line) across all parameter allocations, demonstrating its robust performance advantage. MeSH also shows remarkable parameter efficiency against the non-recursive baseline. As a trend, the performance of `+mesh` (purple line) approaches that of the full 24-layer `Vanilla` model (green dashed line) while using approximately 30% fewer non-embedding parameters. The study shows that MeSH is not just an additive improvement but a powerful architectural principle that enhances the parameter efficiency and scaling properties of recursive transformers.

## 5 CONCLUSION

In this work, we diagnose the underperformance of recursive transformers, tracing it, through the lens of quantified observables, to the systemic pathologies of undifferentiated computation and information overload. We further propose MeSH as a principled architectural solution that externalizes state management into an explicit memory buffer controlled by dynamic, step-wise routers. Our experiments validate that MeSH successfully addresses the diagnosed pathologies while also delivering substantial performance gains on recursive backbones. We conclude that this work establishes explicit, routed state management as a scalable and effective principle for building stronger recursive models, offering a promising architectural path forward as the field seeks more sustainable scaling paradigms.

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

# A    RELATED WORK

**Recursive Transformers and Loop-based LLMs.** The idea of iterating a Transformer layer in a loop originates from the Universal Transformer (UT) (Dehghani et al., 2018), which showed that repeatedly applying a single, weight-shared layer can achieve the expressive power of a much deeper Transformer while allowing variable computation per input. UT also introduced an Adaptive Computation Time mechanism (Graves, 2016) that dynamically adjusts how many iterations to run for each token, together with a trainable positional signal that distinguishes time steps. Since UT, a series of works have extended the concept of looped Transformers (Tan et al., 2023; Giannou et al., 2023; Li & Li, 2021; Takase & Kiyono, 2021; Elbayad et al., 2019; Yang et al., 2023; Zhang et al., 2024; Fan et al., 2024; Hay & Wolf, 2024; Chen et al., 2025b; Nguyen & Lin, 2025; Li et al., 2025a; Aleksandrov et al., 2025; Bae et al., 2025). Recent studies (Saunshi et al., 2025; Zhu et al., 2025b; Geiping et al., 2025) demonstrated—both empirically and theoretically—that increasing depth by looping a small Transformer can match or surpass a far deeper fixed-depth model on challenging reasoning tasks. Later, Zeng et al. (2025) reinforced the view that iterative "pondering" could be critical for test-time scaling and linked the behavior of looped Transformers to an implicit chain-of-thought process. Collectively, these efforts underscore the promise of recursive transformers for adaptive depth and latent reasoning.

**Parameter Sharing and Iteration Differentiation.** A central challenge for recursive transformers is to preserve expressiveness even though all iterations reuse the same parameters. An empirical study of parameter sharing in Transformers showed that naïvely sharing every layer—as in the original UT (Dehghani et al., 2018) —often degrades performance on language tasks, implying that additional mechanisms are required to alleviate representational bottlenecks (Ng & Wang, 2024). Several strategies have been explored: ***Learned loop-index embeddings.*** By injecting a small trainable vector or matrix that encodes the iteration number, models can behave slightly differently at each step while still sharing the main weights (Dehghani et al., 2018; Mohtashami et al., 2023). However, element-wise addition of such embeddings practically produces limited gains (Geiping et al., 2025; Zhu et al., 2025a). ***LoRA per iteration.*** In a similar spirit, recent works (Heo et al., 2025; Bae et al., 2024) attach a separate low-rank adaptation (LoRA) module to each repetition of a pre-trained model, granting every loop its own lightweight set of parameters and mitigating the drawbacks of strict sharing. ***Mixture-of-Experts in a loop.*** MoEUT (Csordás et al., 2024) combines weight sharing with a Mixture-of-Experts (MoE) at every layer: the base layer is reused across iterations, while expert gating adds conditional capacity. MoEUT slightly outperforms a non-loop Transformer of equal compute, underscoring the value of learnable gating and expert routes within a loop architecture. Our work proposes a different paradigm. Instead of adding unique parameters to the loop core to differentiate iterations, we focus on dynamically managing the information flow itself. We propose MeSH to externalize state management into a memory buffer and employ lightweight, step-wise routers to control what is read from and written to it. This externalizes the functional specialization into a routing problem, keeping the core block purely weight-shared and non-invasive.

**Skip Connections and Dense Connectivity.** In deep networks, skip or shortcut connections have long been essential for training very deep architectures effectively (He et al., 2016). Residual Networks (ResNets) and Highway Networks showed that adding identity skip paths improves gradient flow and allows each layer to learn a simpler "update function" on top of an identity mapping He et al. (2016); Srivastava et al. (2015). DenseNet (Huang et al., 2017) further generalized this idea by connecting every layer to all previous layers, so that each layer receives the feature maps of all preceding layers as input; this dense feed-forward architecture promotes feature reuse, mitigates vanishing gradients, and even reduces parameter count. More recently, Hyper-connections (Zhu et al., 2024) widen the hidden state into multiple parallel streams and use learnable coefficients to mix these streams, effectively replacing the standard residual path with a more complex, multi-lane data highway. MUDDformer (Xiao et al., 2025) introduced dense connections into standard decoder-only Transformers, generalizing residuals by adding multiple learnable skip paths between layers; dynamic dense skips allowed a 2.8 B model to match the perplexity of a 6.9 B model with only minimal overhead (Xiao et al., 2025). While conceptually related to dense connectivity, our work focuses on block-to-block (loop-to-loop) connectivity in the recursive setting rather than layer-to-layer wiring. We diverge from direct connections by introducing an external memory buffer and lightweight, step-wise routers. The system facilitates a flexible read-write cycle for managing information flow across iterations, rather than simply gating feed-forward paths. The mechanism is both

principled and more native to the recursive design, as it is explicitly engineered to enable functional specialization between iterations.

**Latent Reasoning and Chain-of-Thought.** Loop-based LLMs are closely related to the idea of latent (hidden) chain-of-thought (Hao et al., 2024; Shen et al., 2025). Instead of explicitly outputting intermediate reasoning steps in natural language, a loop-based model processes those steps internally in vector form (Zhu et al., 2025a; Xu & Sato, 2025). Recent research has examined the differences between prompting a model with an explicit chain-of-thought versus giving it the capacity to "think" silently via latent reasoning (Chen et al., 2025a; Fedorenko et al., 2024; Hao et al., 2024; Pfau et al., 2024). In general, loop offers a promising way to achieve the benefits of multi-step reasoning without incurring the cost of longer outputs or the need for supervised intermediate steps (Li et al., 2025b). Our work contributes to this area by improving the recursive architectural foundation on which latent reasoning unfolds. By enhancing how information is preserved and combined across iterative steps, we aim to make iterative execution more effective. While approaches like CoTFormer inject special tokens to mimic multi-step reasoning inside the model (Mohtashami et al., 2023), most of recursive transformers focus on intrinsic connectivity of loop iterations (Zeng et al., 2025; Geiping et al., 2025; Saunshi et al., 2025). These two methodologies can be seen as complementary, as we can imagine a loop-based LLM that also uses latent CoT training signals. Indeed, analysis of hidden state evolution under different connection schemes can be viewed as an interpretability study of latent reasoning – shedding light on whether the model is gradually refining a solution or oscillating, and how much it relies on initial information versus newly computed results at each step.

**Memory-Augmented Transformers.** Our work is situated within the broad field of memory-augmented Transformers (Omidi et al., 2025), including diverse approaches to enhance neural networks with memory modules. A historic line of memory-augmented neural networks use memory for long-term knowledge storage and algorithmic reasoning, evolving from the cell state in LSTMs (Hochreiter & Schmidhuber, 1997) to architectures with explicit external memory matrices, such as the Neural Turing Machine (Graves et al., 2014) and its sparse-access variants (Rae et al., 2016). In the Transformer era, a line of work uses memory to address the fixed context-length limitation of Transformers. This began with Transformer-XL (Dai et al., 2019) caching hidden states from previous segments, was improved by Compressive Transformer (Rae et al., 2020) which compresses older states, and was further developed by models like RMT (Bulatov et al., 2022) and Memformer (Wu et al., 2020) that introduce dedicated memory tokens or slots. More recent models, such as LONG-MEM (Wang et al., 2023) and the Memorizing Transformer (Wu et al., 2022), employ retrieval-based augmentation, where key-value pairs from past segments are stored in an external bank and a retrieval mechanism is used to pull relevant information into the current context. The shared objective of these varied approaches is to enable information flow across long temporal sequences or distinct, sequential input segments for knowledge storage or context extension. In contrast, our proposed MeSH mechanism operates on a single, fixed-length input, where the memory buffer manages the intermediate hidden states generated during successive computational iterations over that same input. The function of MeSH is to structure the information flow within a token-wise recursive computation, a target distinct from long-term knowledge storage or long-context extension.

## B    EXPERIMENTAL DETAILS

**Pre-training.** All models are pretrained from scratch, closely following the methodology of the Pythia suite (Biderman et al., 2023). Our training is conducted on a 250B-token deduplicated subset of the Pile dataset (Gao et al., 2020), using the original GPT-NeoX tokenizer with a vocabulary size of 50,257. All models are trained for one epoch.

**Model Architecture.** Our implementations are based on the GPT-NeoX architecture provided by the Pythia suite (Biderman et al., 2023). For recursive models, we adopt the prelude-loop-coda structure. We denote the layer distribution using the notation $L_{pre}+L_{core}RN_{loop}+L_{coda}$. For example, a `4+8R2+4` configuration corresponds to a model with a 4-layer prelude ($L_{pre}$), an 8-layer shared core ($L_{core}$) that is looped twice ($N_{loop}$), and a 4-layer coda ($L_{coda}$). Our `+mesh` variant is implemented by inserting a state buffer and step-wise routers at the boundaries of these conceptual blocks. Each router consists of a single linear layer followed by a softmax function to generate dynamic routing weights. The buffer size $B$ is set following the empirical $B = N_{loop} + 3$ derived from our ablation study (see Appendix E.2). Following standard Transformer practice (Vaswani et al.,

2017), we scale the input embeddings by a factor of $\sqrt{d_{\text{model}}}$ before they enter the first layer. To ensure training stability in these deep computational graphs, we employ a depth-aware weight initialization, scaling the standard deviation of output projection weights by $1/\sqrt{2 \times N_{\text{compute}}}$, where $N_{\text{compute}}$ is the total number of layers in the unrolled computation graph.

**Training Hyperparameters.** We use the AdamW optimizer with $\beta_1 = 0.9$, $\beta_2 = 0.95$, and a weight decay of $0.01$. The learning rate follows a cosine decay schedule with a 1% warmup, decaying to 10% of the peak value. The peak learning rate is scaled according to model size, ranging from $6.0 \times 10^{-4}$ for the 160M model to $2.0 \times 10^{-4}$ for the 1.4B model. All models are trained with a consistent global batch size of 512 and a sequence length of 4096 tokens. To improve training efficiency, we utilize BF16 mixed-precision and FlashAttention-2 (Dao, 2023). Our distributed training setup is managed by DeepSpeed with ZeRO Stage 0.

**Downstream Task Evaluation.** To assess model performance, we evaluated few-shot accuracy on 9 benchmarks using the Language Model Evaluation Harness framework (Gao et al., 2024). The evaluation suite includes: Lambada (Paperno et al., 2016) in both its OpenAI (LD-O) and Standard (LD-S) versions, PIQA (PQ) (Bisk et al., 2020), HellaSwag (HS) (Zellers et al., 2019), WinoGrande (WG) (Sakaguchi et al., 2021), ARC-Easy (ARC-E) and ARC-Challenge (ARC-C) (Clark et al., 2018), SciQ (Welbl et al., 2017), and continuation-MMLU (cMMLU) (Hendrycks et al., 2020). We report accuracy normalized by the byte length of the target string for PIQA, HellaSwag, ARC-E, ARC-C, and SciQ and standard accuracy for Lambada, WinoGrande, and cMMLU. All evaluations are conducted in both *0-shot* and *5-shot* settings. All measurements were performed on a single NVIDIA H20 GPU. Detailed results are shown in Table 2.

## C  PSEUDOCODE

We provide detailed pseudocode for the recursive architectures discussed in the main paper. Algorithm 1 outlines the implementation of common recursive variants, which rely on fixed, heuristic-based state-passing schemes. In contrast, Algorithm 2 details our proposed MeSH-augmented recurrence, which replaces the rigid logic with a dynamic, memory-based system.

---

**Algorithm 1** Recursive Transformers with Common Variants

---

1: **Input:** Token embeddings $\mathbf{h}_{\text{emb}}$, Prelude $f_{\text{pre}}$, Core $f_{\text{core}}$, Coda $f_{\text{coda}}$
2: **Hyperparameters:** Loop iterations $K$, Variant type $\in \{\texttt{base}, \texttt{residual}, \texttt{anchor}\}$

3: # 1. Prelude
4: $\mathbf{h}^{(0)} \leftarrow f_{\text{pre}}(\mathbf{h}_{\text{emb}})$ {Compute initial state for the loop}

5: # 2. Main Recursive Loop
6: **for** $t = 0$ **to** $K-1$ **do do**
7:     # — Select supplementary state based on variant —
8:     $\mathbf{h}_{\text{sup}}^{(t)} \leftarrow 0$
9:     **if** Variant type is $\texttt{residual}$ **then**
10:         $\mathbf{h}_{\text{sup}}^{(t)} \leftarrow \mathbf{h}^{(t)}$
11:     **else if** Variant type is $\texttt{anchor}$ **then**
12:         $\mathbf{h}_{\text{sup}}^{(t)} \leftarrow \mathbf{h}^{(0)}$
13:     **else if** Variant type is $\texttt{anchor}\star$ **then**
14:         $\mathbf{h}_{\text{sup}}^{(t)} \leftarrow \mathbf{h}_{\text{emb}}$
15:     **end if**
16:     $\mathbf{h}^{(t+1)} \leftarrow f_{\text{core}}(\mathbf{h}^{(t)}) + \mathbf{h}_{\text{sup}}^{(t)}$ {Apply core and add supplement}
17: **end for**

18: # 3. Final Coda Processing
19: $\mathbf{h}_{\text{final}} \leftarrow f_{\text{coda}}(\mathbf{h}^{(K)})$
20: **return** $\mathbf{h}_{\text{final}}$

---

Table 2: Detailed downstream evaluation results (stacked). For each model variant, performance is shown for both *0-shot* and *5-shot* settings. We report accuracy values for all tasks. The average accuracy ("Avg.") is computed over the 9 preceding tasks. Dataset abbreviations correspond to: `LD-O` (Lambda OpenAI), `LD-S` (Lambda Standard), `HS` (HellaSwag), `PQ` (PIQA), `WG` (WinoGrande), `ARC-E` (ARC-easy), `ARC-C` (ARC-Challenge), `SciQ` (SciQ), and `cMMLU` (MMLU-continuation).

| | Structure | | | | Downstream Task Performance | | | | | | | | | |
| | Scheme | Layers | Variant | | LD-O | LD-S | HS | PQ | WG | ARC-E | ARC-C | SciQ | cMMLU | Avg. |
|---|---|---|---|---|---|---|---|---|---|---|---|---|---|---|
| **Pythia-160M** | Vanilla | 12 | — | *0-shot* | 32.31 | 23.64 | 31.14 | 62.46 | 50.59 | 39.56 | 23.21 | 70.3 | 25.69 | 39.88 |
| | | | | *5-shot* | 27.11 | 24.22 | 31.38 | 62.95 | 50.67 | 42.21 | 22.53 | 78.2 | 25.55 | 40.54 |
| | Recursive (-33.3%) | 2+4R2+2 | base | *0-shot* | 29.30 | 20.18 | 30.85 | 60.72 | 49.57 | 40.03 | 23.21 | 70.9 | 25.30 | 38.90 |
| | | | | *5-shot* | 24.32 | 19.43 | 30.76 | 61.43 | 51.14 | 41.75 | 22.53 | 76.5 | 25.75 | 39.29 |
| | | | +anchor | *0-shot* | 30.04 | 21.11 | 30.93 | 60.39 | 51.14 | 38.13 | 23.81 | 68.3 | 25.40 | 38.81 |
| | | | | *5-shot* | 26.16 | 21.23 | 31.44 | 61.15 | 50.75 | 41.71 | 23.04 | 80.2 | 25.70 | 40.15 |
| | | | +mesh | *0-shot* | 31.32 | 21.48 | 31.02 | 60.66 | 53.43 | 39.06 | 22.27 | 69.7 | 25.73 | 39.41 |
| | | | | *5-shot* | 26.43 | 21.00 | 31.48 | 60.72 | 51.93 | 42.93 | 23.04 | 81.9 | 26.00 | 40.60 |
| **Pythia-410M** | Vanilla | 24 | — | *0-shot* | 41.74 | 29.65 | 37.65 | 64.80 | 51.93 | 43.60 | 25.68 | 73.1 | 26.68 | 43.87 |
| | | | | *5-shot* | 35.59 | 28.92 | 38.01 | 67.19 | 50.08 | 50.08 | 25.43 | 85.2 | 27.03 | 45.31 |
| | Recursive (-33.3%) | 4+8R2+4 | base | *0-shot* | 39.47 | 30.43 | 36.71 | 63.71 | 53.59 | 42.59 | 24.57 | 71.8 | 26.47 | 43.26 |
| | | | | *5-shot* | 35.22 | 28.26 | 36.71 | 64.91 | 52.88 | 48.86 | 25.77 | 85.7 | 26.95 | 45.03 |
| | | | +anchor | *0-shot* | 41.45 | 31.24 | 36.82 | 64.09 | 52.80 | 43.14 | 23.72 | 73.6 | 26.39 | 43.70 |
| | | | | *5-shot* | 36.56 | 30.06 | 37.13 | 65.62 | 51.30 | 49.24 | 25.43 | 88.8 | 26.94 | 45.68 |
| | | | +mesh | *0-shot* | 41.92 | 32.33 | 37.27 | 64.20 | 53.83 | 42.30 | 25.09 | 73.5 | 26.66 | 44.12 |
| | | | | *5-shot* | 36.25 | 31.71 | 38.00 | 65.40 | 51.22 | 49.37 | 24.49 | 87.0 | 26.93 | 45.56 |
| | Recursive (-50.0%) | 3+6R3+3 | base | *0-shot* | 37.32 | 25.93 | 35.43 | 63.44 | 50.67 | 41.79 | 23.38 | 72.8 | 26.68 | 41.94 |
| | | | | *5-shot* | 30.97 | 25.64 | 35.95 | 64.96 | 51.93 | 47.73 | 24.40 | 87.7 | 26.77 | 44.01 |
| | | | +residual | *0-shot* | 37.80 | 27.98 | 35.60 | 64.64 | 52.01 | 42.30 | 23.98 | 68.7 | 26.42 | 42.16 |
| | | | | *5-shot* | 33.13 | 28.45 | 35.65 | 65.29 | 50.20 | 47.39 | 25.00 | 86.3 | 26.78 | 44.24 |
| | | | +anchor | *0-shot* | 38.33 | 29.11 | 36.01 | 65.45 | 51.46 | 43.18 | 22.78 | 72.8 | 26.56 | 42.85 |
| | | | | *5-shot* | 33.92 | 29.44 | 36.61 | 65.56 | 53.04 | 47.22 | 23.89 | 87.8 | 26.65 | 44.90 |
| | | | +mesh | *0-shot* | 41.88 | 31.87 | 36.86 | 65.51 | 52.17 | 42.26 | 24.15 | 70.9 | 26.15 | 43.53 |
| | | | | *5-shot* | 37.84 | 31.85 | 37.08 | 65.34 | 53.20 | 48.82 | 25.60 | 87.8 | 26.79 | 46.04 |
| **Pythia-1B** | Vanilla | 16 | — | *0-shot* | 46.73 | 34.02 | 43.61 | 66.87 | 52.01 | 48.53 | 26.28 | 76.6 | 27.86 | 46.95 |
| | | | | *5-shot* | 40.60 | 34.41 | 43.98 | 68.44 | 52.33 | 54.46 | 28.75 | 89.9 | 28.71 | 49.07 |
| | Recursive (-31.3%) | 3+5R2+3 | base | *0-shot* | 45.76 | 33.84 | 41.57 | 66.87 | 52.25 | 45.83 | 25.77 | 72.0 | 27.55 | 45.72 |
| | | | | *5-shot* | 38.31 | 31.21 | 42.54 | 68.12 | 52.96 | 53.41 | 26.62 | 89.1 | 28.33 | 47.75 |
| | | | +residual | *0-shot* | 45.70 | 34.06 | 41.85 | 66.49 | 52.49 | 47.10 | 26.02 | 74.4 | 27.61 | 46.19 |
| | | | | *5-shot* | 38.15 | 32.78 | 42.50 | 67.52 | 53.43 | 52.78 | 26.11 | 89.0 | 28.35 | 47.85 |
| | | | +anchor* | *0-shot* | 47.62 | 35.15 | 43.06 | 67.25 | 53.35 | 46.51 | 25.68 | 75.1 | 27.97 | 46.85 |
| | | | | *5-shot* | 42.46 | 34.58 | 43.24 | 68.55 | 52.01 | 55.22 | 27.82 | 90.3 | 28.42 | 49.18 |
| | | | +anchor | *0-shot* | 46.17 | 34.68 | 42.62 | 67.68 | 53.51 | 46.80 | 25.26 | 75.9 | 27.99 | 46.73 |
| | | | | *5-shot* | 39.92 | 32.89 | 43.26 | 69.15 | 53.12 | 55.05 | 27.22 | 90.0 | 28.83 | 48.83 |
| | | | +mesh | *0-shot* | 48.40 | 36.95 | 44.36 | 67.03 | 52.01 | 46.93 | 26.54 | 77.6 | 27.91 | 47.53 |
| | | | | *5-shot* | 42.62 | 34.87 | 44.68 | 67.95 | 52.96 | 55.14 | 27.22 | 91.4 | 28.71 | 49.51 |
| **Pythia-1.4B** | Vanilla | 24 | — | *0-shot* | 51.08 | 39.82 | 47.74 | 68.83 | 55.41 | 50.04 | 26.11 | 77.3 | 29.18 | 49.50 |
| | | | | *5-shot* | 46.17 | 39.69 | 48.01 | 69.64 | 54.22 | 59.22 | 29.27 | 91.2 | 29.95 | 51.93 |
| | Recursive (-33.3%) | 4+8R2+4 | base | *0-shot* | 49.56 | 39.32 | 46.50 | 69.37 | 53.67 | 49.79 | 27.56 | 75.9 | 28.34 | 48.89 |
| | | | | *5-shot* | 44.95 | 38.04 | 46.73 | 69.59 | 54.30 | 56.82 | 28.58 | 90.4 | 29.52 | 50.99 |
| | | | +residual | *0-shot* | 51.08 | 41.10 | 47.10 | 69.04 | 53.12 | 49.03 | 26.79 | 79.6 | 28.66 | 49.50 |
| | | | | *5-shot* | 47.20 | 38.81 | 47.06 | 69.26 | 54.30 | 56.23 | 27.65 | 90.9 | 29.18 | 51.18 |
| | | | +anchor* | *0-shot* | 51.35 | 41.28 | 47.28 | 67.90 | 55.09 | 49.16 | 27.47 | 75.3 | 28.76 | 49.29 |
| | | | | *5-shot* | 45.88 | 40.17 | 47.72 | 69.31 | 53.75 | 58.25 | 28.75 | 93.0 | 29.61 | 51.83 |
| | | | +anchor | *0-shot* | 50.75 | 40.93 | 47.65 | 69.75 | 53.75 | 48.40 | 26.62 | 78.0 | 28.62 | 49.39 |
| | | | | *5-shot* | 45.99 | 40.95 | 47.85 | 69.37 | 52.96 | 56.82 | 26.79 | 91.0 | 29.65 | 51.27 |
| | | | +mesh | *0-shot* | 53.46 | 41.84 | 48.58 | 69.53 | 54.85 | 49.75 | 27.82 | 80.3 | 28.89 | 50.56 |
| | | | | *5-shot* | 49.14 | 42.69 | 49.21 | 69.70 | 54.78 | 57.79 | 29.35 | 92.7 | 29.76 | 52.79 |

# D DISCUSSION: EXPRESSIVE POWER OF MESH AS A GENERAL RECURRENCE

The baseline recurrences described in Section 3.1 employ a fixed, non-adaptive state update rule: the output of the core block, $\mathbf{h}_m^{(t)} = f_{\text{core}}(\mathbf{h}^{(t)})$, is always supplemented by a predetermined state (e.g., zero, the previous state $\mathbf{h}^{(t)}$, or the initial state $\mathbf{h}^{(0)}$). We propose that MeSH offers a more general and powerful alternative by replacing this rigid addition with a learnable, dynamic state composition mechanism.

---

**Algorithm 2** MeSH-Augmented Recurrence within a Prelude-Recurrent-Coda Structure

---

1: **Input:** Token embeddings $\mathbf{h}_{\text{emb}}$, Prelude $f_{\text{pre}}$, Core $f_{\text{core}}$, Coda $f_{\text{coda}}$
2: **Parameters:** MeSH buffer $\mathbf{M}$, Routers $\{R_{\text{write}}^{(t)}, R_{\text{read}}^{(t)}\}_{t=-1}^{K-1}$
3: **Hyperparameters:** Loop iterations $K$, Buffer slots $B$

4: # 1. Initialize MeSH Buffer
5: $\mathbf{M}^{(0)} \leftarrow$ zeros {Initialize buffer with zeros}
6: $\mathbf{m}_0^{(0)} \leftarrow \mathbf{h}_{\text{emb}}$ {Place embeddings in the first slot}

7: # 2. Prelude
8: $\mathbf{h}_{\text{m}}^{(-1)} \leftarrow f_{\text{pre}}(\mathbf{h}_{\text{emb}})$ {Compute prelude output}
9: $\mathbf{w}_{\text{write}}^{(-1)}, \mathbf{w}_{\text{read}}^{(-1)} \leftarrow \text{Routers}^{(t=-1)}(\mathbf{h}_{\text{m}}^{(-1)})$ {Use transitional routers}
10: **for** $b = 0$ **to** $B - 1$ **do do**
11:     $\mathbf{m}_b^{(0)} \leftarrow \mathbf{m}_b^{(0)} + \mathbf{h}_{\text{m}}^{(-1)} \odot \mathbf{w}_{\text{write},b}^{(-1)}$ {Write prelude output to buffer}
12: **end for**
13: $\mathbf{h}^{(0)} \leftarrow \sum_{b=0}^{B-1} \mathbf{m}_b^{(0)} \odot \mathbf{w}_{\text{read},b}^{(-1)}$ {Synthesize first loop state}

14: # 3. Main Recursive Loop
15: **for** $t = 0$ **to** $K - 1$ **do do**
16:     $\mathbf{h}_{\text{m}}^{(t)} \leftarrow f_{\text{core}}(\mathbf{h}^{(t)})$ {Core computation}
17:     $\mathbf{w}_{\text{write}}^{(t)}, \mathbf{w}_{\text{read}}^{(t)} \leftarrow \text{Routers}^{(t)}(\mathbf{h}^{(t)})$ {Compute step-wise weights}
18:     $\mathbf{M}^{(t+1)} \leftarrow \mathbf{M}^{(t)}$
19:     **for** $b = 0$ **to** $B - 1$ **do do**
20:         $\mathbf{m}_b^{(t+1)} \leftarrow \mathbf{m}_b^{(t+1)} + \mathbf{h}_{\text{m}}^{(t)} \odot \mathbf{w}_{\text{write},b}^{(t)}$ {Update buffer with a distributed write}
21:     **end for**
22:     $\mathbf{h}^{(t+1)} \leftarrow \sum_{b=0}^{B-1} \mathbf{m}_b^{(t+1)} \odot \mathbf{w}_{\text{read},b}^{(t)}$ {Synthesize next state}
23: **end for**

24: # 4. Final Coda Processing
25: $\mathbf{h}_{\text{final}} \leftarrow f_{\text{coda}}(\mathbf{h}^{(K)})$ {Use the state after the last read}
26: **return** $\mathbf{h}_{\text{final}}$

---

**Proposition 2.1.** *The MeSH recurrence, defined by the compute-write-read cycle, generalizes the concept of additive state updates (as in residual and anchor variants) by learning to dynamically retrieve and combine historical states from memory to form the state for the next iteration.*

**Demonstration.** To reveal the underlying mechanics, we can unroll the MeSH update equations. The state for the next iteration, $\mathbf{h}^{(t+1)}$, is formed by reading from the just-updated memory $\mathbf{M}^{(t+1)}$:

$$
\begin{aligned}
\mathbf{h}^{(t+1)} &= \sum_{b=0}^{B-1} \mathbf{m}_b^{(t+1)} \odot \mathbf{w}_{\text{read},b}^{(t)} \\
&= \sum_{b=0}^{B-1} \left( \mathbf{m}_b^{(t)} + \mathbf{h}_{\text{m}}^{(t)} \odot \mathbf{w}_{\text{write},b}^{(t)} \right) \odot \mathbf{w}_{\text{read},b}^{(t)} \\
&= \underbrace{\sum_{b=0}^{B-1} \mathbf{m}_b^{(t)} \odot \mathbf{w}_{\text{read},b}^{(t)}}_{\text{Retrieved Historical Summary}} + \underbrace{\left( \sum_{b=0}^{B-1} \mathbf{w}_{\text{write},b}^{(t)} \odot \mathbf{w}_{\text{read},b}^{(t)} \right)}_{\text{Gating Factor}} \odot \mathbf{h}_{\text{m}}^{(t)}
\end{aligned}
\tag{6}
$$

Let us analyze the two resulting components. The second term is the core's output, $\mathbf{h}_{\text{m}}^{(t)}$, scaled by a learned gating factor. The first term is a dynamic retrieval of information from the memory state $\mathbf{m}^{(t)}$ *before* the current write operation. Note that this term is distinct from the previous state $\mathbf{h}^{(t)}$, which was formed using the read weights from the prior step, $\mathbf{w}_{\text{read}}^{(t-1)}$.

The formulation reveals that the next state $\mathbf{h}^{(t+1)}$ is a generalized residual update composed of:

1. A **retrieved historical summary** that dynamically combines states presented in the buffer. The read router learns what historical information is most relevant at this step.

2. A **gated output** of the current core block, $\mathbf{h}_{\mathrm{m}}^{(t)}$, scaled by a learned gating factor.

The dynamic process generalizes the fixed baseline recurrences, as we can conceptually unroll the **retrieved historical summary** even further as a weighted combination of the initial memory state and all previous core outputs $\{\mathbf{h}_{\mathrm{m}}^{(0)}, \ldots, \mathbf{h}_{\mathrm{m}}^{(t-1)}\}$ that have been written to the buffer. Therefore, the next state $\mathbf{h}^{(t+1)}$ can be viewed as a comprehensive, dynamic aggregation of all computations performed so far:

$$\mathbf{h}^{(t+1)} = \alpha_t \odot \mathbf{h}_{\mathrm{m}}^{(t)} + \sum_{i=-1}^{t-1} \alpha_i \odot \mathbf{h}_{\mathrm{m}}^{(i)} + \alpha_{\mathrm{emb}} \odot \mathbf{h}_{\mathrm{emb}} \tag{7}$$

where all coefficients $\alpha$ are implicit coupled with write and read weights during previous iterations. The perspective makes the generalization self-evident:

- **Simulating anchor ($\mathbf{h}_{\mathbf{m}}^{(t)} + \mathbf{h}_{\mathbf{emb}}$):** This is achieved by learning a routing scheme where the coefficients in Eq. 7 are set as follows: the weight for the current computation, $\alpha_t$, approaches 1; the weight for the initial embedding, $\alpha_{\mathrm{emb}}$, approaches 1; and all other historical weights, $\alpha_{i<t}$, are driven to zero. MeSH can learn to adopt this specific weighting only when needed, rather than being hard-wired to it.

- **Simulating residual ($\mathbf{h}_{\mathbf{m}}^{(t)} + \mathbf{h}^{(t)}$):** To approximate this, MeSH needs to reconstruct the previous state, $\mathbf{h}^{(t)}$, as its historical summary. This is naturally achievable. Since $\mathbf{h}^{(t)}$ is itself a weighted sum of $\{\mathbf{h}_{\mathrm{emb}}, \mathbf{h}_{\mathrm{m}}^{(0)}, \ldots, \mathbf{h}_{\mathrm{m}}^{(t-1)}\}$, the routers at step $t$ can learn to compute the appropriate coefficients ($\alpha_{\mathrm{emb}}, \alpha_0, \ldots, \alpha_{t-1}$) to reconstruct or closely approximate $\mathbf{h}^{(t)}$. More powerfully, MeSH can choose to form a "better" historical summary by up-weighting more relevant past states (e.g., $\mathbf{h}_{\mathrm{m}}^{(t-5)}$) and down-weighting irrelevant ones (e.g., $\mathbf{h}_{\mathrm{m}}^{(t-1)}$), thus forming more effective long-range dependencies.

- **Adaptive Combination:** The core advantage is that the coefficients $\alpha$ are not fixed. They are functions of the current state $\mathbf{h}^{(t)}$, allowing the model to change its recurrence rule on the fly. It can learn to behave like an Anchor in early steps, transition to a Residual-like update, and synthesize a complex summary from multiple past states for the final output, all within a single forward pass.

In conclusion, MeSH does not just replicate the fixed recurrences; it subsumes the underlying principle of combining past and present information into a flexible, learnable framework. It replaces the hard-coded "what to add" (e.g., $\mathbf{h}^{(0)}$ or $\mathbf{h}^{(t)}$) with a learned "what to retrieve and combine," offering a substantially more expressive mechanism for managing state in recursive transformers.

## E    MORE RESULTS

### E.1    SCALING TO LARGER RECURSIVE MODELS

To further investigate the scalability of MeSH, we conducted additional experiments on larger Pythia models, specifically at the 2.8B and 6.9B scales. For these models, we employed a recursive configuration of `6+10R2+6`, which achieves a non-embedding parameter reduction to 68.75% compared to their non-recursive counterparts. We compared the performance of the base recursive model (`rec`) against the MeSH-enhanced version (`+mesh`) in Table 3. The results show that the benefits of MeSH scale effectively to larger models. At the 2.8B and 6.9B scales, MeSH still delivers substantial improvements across the board, significantly lowering perplexity on all four datasets and boosting 0-shot and 5-shot average accuracy. This scaling experiment provides strong evidence that MeSH is a robust and effective method for enhancing the performance of recursive language models at multi-billion parameter scales.

Table 3: **Performance of MeSH on larger-scale models (Pythia-2.8B and 6.9B).** Both models use a `6+10R2+6` recursive configuration. Since no vanilla counterparts were trained, $\Delta$acc for the `+mesh` variant shows the absolute accuracy change relative to its `base` recursive counterpart. Best results within each model block are **bolded**.

| | Structure | | | Perplexity↓ | | | | Task Avg. acc↑ | |
|---|---|---|---|---|---|---|---|---|---|
| | Scheme | Layers | Variant | Pile | Wiki | LD-O | LD-S | 0-shot | 5-shot |
| Pythia-2.8B | Recursive (-31.25%) | 6+10R2+6 | base | 6.90 | 14.18 | 8.41 | 16.87 | 52.49 | 54.92 |
| | | | +mesh | **6.70** | **13.60** | **7.30** | **11.36** | **54.71** | **56.85** |
| Pythia-6.9B | Recursive (-31.25%) | 6+10R2+6 | base | 6.29 | 12.14 | 6.34 | 10.29 | 56.67 | 59.43 |
| | | | +mesh | **6.09** | **11.64** | **5.48** | **8.66** | **58.83** | **60.49** |

Table 4: Ablation on MeSH buffer length across multiple datasets for the Pythia-410M recursive model (`4+8R2+4`). Performance is measured in perplexity ($\downarrow$).

| Scratchpad Slots ($k$) | Buffer Length $\left(B = (N_{\text{loop}} + 1) + k\right)$ | Perplexity↓ | | | |
|---|---|---|---|---|---|
| | | **Pile** | **Wiki** | **LD-O** | **LD-S** |
| 0 | $(2+1) + 0 = 3$ | 9.1231 | 21.9348 | 20.8286 | 55.9474 |
| 1 | $(2+1) + 1 = 4$ | 9.1003 | 21.8439 | 20.6861 | 48.9034 |
| 2 | $(2+1) + 2 = 5$ | **9.0944** | **21.8351** | **19.6316** | **42.5129** |
| 3 | $(2+1) + 3 = 6$ | 9.1088 | 21.9120 | 19.7172 | 56.5541 |

## E.2 ABLATION STUDY: MeSH BUFFER LENGTH

To establish a principled heuristic for setting the MeSH buffer length ($B$), we hypothesize that its capacity should scale with the number of major computational states generated during the recursive process. For a model with $N_{\text{loop}}$ iterations, this includes the initial state from the prelude network plus the output from each of the $N_{\text{loop}}$ core blocks, totaling $N_{\text{loop}} + 1$ essential states. We therefore model the buffer size as $B = (N_{\text{loop}} + 1) + k$, where $k$ is the number of auxiliary "scratchpad" slots available for flexible composition.

We conduct an ablation study to find the optimal $k$ using the Pythia-410M model with a `4+8R2+4` configuration, where $N_{\text{loop}} = 2$, evaluating the performance across four the evaluation datasets. The results are shown in Table 4. Performance improves as we add scratchpad slots, peaking at $k = 2$. This configuration, corresponding to a total buffer length of $B = 5$, achieves the lowest perplexity on all four datasets. Performance slightly degrades at $k = 3$, suggesting a point of diminishing returns. This indicates a sweet spot where the buffer has dedicated slots for each major computational state, plus two auxiliary slots for managing intermediate representations, without making the routing task overly complex. Based on empirical results, we adopt the general rule $B = N_{\text{loop}} + 3$ for all MeSH models in our main experiments.

## E.3 ABLATION STUDY: HEURISTIC STATE-PASSING SCHEMES

While individual heuristic schemes like `+anchor` and `+residual` improve over the base recursive model, a natural question arises: can we achieve further gains by combining them, and can the model learn the optimal combination? To investigate this, we conducted an ablation study on the Pythia-410M model (4+8R2+4 configuration) exploring both fixed additive combinations and learnable linear combinations of supplementary states ($\mathbf{h}_{\text{sup}}^{(t)}$). For the learnable schemes, we define the supplementary state as a combination of the base, anchor, anchor*, and residual states: $\mathbf{h}^{(t)} = \alpha_1 \mathbf{h}_{\text{m}}^{(t)} + \alpha_2 \mathbf{h}^{(0)} + \alpha_3 \mathbf{h}_{\text{emb}}$. We test two variants for the coefficients $\alpha_i$:

- **Static Combination:** The coefficients are trainable scalar parameters that are fixed after training (Ng & Wang, 2024).
- **Dynamic Combination:** The coefficients are dynamically computed at each iteration based on the previous state $\mathbf{h}^{(t-1)}$.

The results, summarized in Table 5, reveal the brittleness of heuristic design. Simply adding all states (+residual+anchor+anchor*) degrades performance, yielding a higher perplexity (9.24) than using +anchor alone (9.19). This confirms that a naive "more is better" approach is not a reliable strategy. While a carefully hand-picked combination (+anchor+anchor*) achieves the best result among all explicit schemes (9.17 PPL), this requires manual tuning. The learnable static and dynamic combinations effectively avoid the worst-case performance degradation but fail to match the best-performing heuristic.

Table 5: Ablation on combinations of supplementary context schemes for Pythia-410M (4+8R2+4). We report evaluation loss and perplexity on the Pile dataset. While a hand-picked combination (+anchor+anchor*) works best among heuristics, MeSH surpasses all explicit schemes.

| Scheme | Loss ↓ | PPL ↓ |
|---|---|---|
| vanilla | 2.2047 | 9.0675 |
| recursive-base | 2.2358 | 9.3542 |
| *Single Heuristic Baselines* | | |
| +anchor | 2.2178 | 9.1867 |
| *Fixed Additive Combinations* | | |
| +residual+anchor | 2.2251 | 9.2541 |
| +residual+anchor+anchor* | 2.2237 | 9.2415 |
| +anchor+anchor* | 2.2159 | 9.1694 |
| *Learnable Combinations* | | |
| +static comb. | 2.2163 | 9.1731 |
| +dynamic comb. | 2.2176 | 9.1851 |
| +mesh (ours) | **2.2077** | **9.0944** |

This suggests that while explicit, learnable weighting can provide a "safe" baseline by ignoring detrimental combinations, it lacks the expressive capacity to discover optimal synergistic interactions. In sharp contrast, our +mesh model (9.09 PPL) significantly outperforms all heuristic-based approaches. Instead of being constrained to an explicit, low-dimensional linear combination of predefined states, MeSH learns a complex, non-linear function for retrieving and composing information from its memory buffer. This allows it to discover implicit, high-dimensional combinations, effectively breaking through the performance ceiling imposed by simpler, explicit state-passing schemes.

## E.4 COMPLEXITY ANALYSIS

Table 6: Parameter counts for Pythia-1.4B variants. Percentages show the change relative to the vanilla baseline for total and non-embedding parameters.

| Model | Variant | Config | Total Params | Non-Embedding Params | Router Weights |
|---|---|---|---|---|---|
| Pythia-1.4B | vanilla | 24 | 1,423,036,416 | 1,208,602,624 | / |
| | recursive | 4+8R2+4 | 1,020,170,240 (-28.310%) | 805,736,448 (-33.333%) | / |
| | +mesh | 4+8R2+4 ($B = 5$) | 1,020,231,710 (-28.306%) | 805,797,918 (-33.328%) | 61,470 (+0.005%) |

**Parameter Overhead.** MeSH introduces a set of lightweight, step-wise routers for its read and write operations. The total number of additional parameters is determined by $(N_{\text{loop}} + 1) \times D_{\text{hidden}} \times B \times 2$, where $N_{\text{loop}}$ is the number of loop iterations, $D_{\text{hidden}}$ is the hidden size, $B$ is the buffer length, and the factor of 2 accounts for both read and write routers. This overhead is negligible compared to the significant parameter savings achieved through recursion. As detailed in Table 6, for our Pythia-1.4B model in a 4+8R2+4 configuration, the recursive structure reduces the non-embedding parameters by 33.33% compared to its vanilla counterpart. The MeSH routers add a

mere 61,470 parameters (0.005% relative to the non-embedding part), which shows that MeSH achieves its substantial performance gains with virtually no cost to parameter efficiency, making it an architecturally lightweight yet powerful enhancement.

Table 7: **FLOPs overhead analysis for Pythia-1.4B recursive variants.** The total GFLOPs are measured for a single forward pass with an input of size [1, 4096]. The overhead is calculated relative to the base recursive model.

| Model | Variant | Config | Total GFLOPs (1e9) | Extra GFLOPs (1e9) |
|---|---|---|---|---|
| Pythia-1.4B | recursive (base) | 4+8R2+4 | 5373.792 | / |
| | +residual | 4+8R2+4 | 5373.809 | 0.0168 (+0.000312%) |
| | +anchor | 4+8R2+4 | 5373.809 | 0.0168 (+0.000312%) |
| | +mesh | 4+8R2+4 ($B = 5$) | 5374.547 | 0.7551 (+0.014051%) |

**Computational Overhead.** To quantify the computational cost, we measured the FLOPs for a single forward pass using the `fvcore` library on our Pythia-1.4B model with a `4+8R2+4` recursive configuration and a standard input tensor of shape [1, 4096]. We report the computational overhead of the `+residual`, `+anchor` and `+mesh` variants relative to the `base` recursive model in Table 7. The analysis reveals that the overhead from MeSH is negligible, adding only approximately +0.014% for a single forward pass. This efficiency is theoretically grounded. Let bs be the batch size, $S_{len}$ the sequence length, $D_{hidden}$ the hidden dimension, $N_{loop}$ the number of loops, and $B$ the buffer length. The total extra FLOPs for MeSH are given by:

$$\Delta \text{FLOPs}_{\text{mesh}} = (N_{loop} + 1) \cdot (6 \cdot \text{bs} \cdot S_{len} \cdot D_{hidden} \cdot B)$$ (8)

For our configuration ($N_{loop} = 2, \text{bs} = 1, S_{len} = 4096, D_{hidden} = 2048, B = 5$), this formula yields $\approx 0.755$ GFLOPs, matching the empirically measured overhead. The analysis confirms that the significant performance and stability gains provided by MeSH are achieved with a minimal and practically insignificant increase in computational requirements, highlighting its architectural efficiency.

### E.5 DETAILED TRAINING DYNAMICS ON DOWNSTREAM TASKS

To provide a more granular view of the training dynamics presented in Section 4.3, Figure 9 shows the performance of the 1.4B-parameter models on 9 individual downstream tasks and their average accuracy, evaluated at various checkpoints throughout the pre-training process.

### E.6 APPLYING MESH TO NON-RECURSIVE ARCHITECTURES

In our main experiments, the `recursive+mesh` model for Pythia-1.4B surpasses its larger `Vanilla` counterpart, even with 33.3% fewer parameters. The result suggests that the MeSH mechanism might offer architectural benefits beyond the the recursive setting. We hypothesize that if the performance bottleneck of parameter sharing were removed, the benefits of MeSH could be even more pronounced. We conduct an experiment applying a MeSH-like structure to a standard, non-recursive `Vanilla` transformer. We conceptually partition the 24 layers of the Pythia-1.4B `Vanilla` model into blocks that mirror our `4+8R2+4` recursive design: a 4-layer prelude, two distinct 8-layer core blocks (`core_1` and `core_2`), and a 4-layer coda. Crucially, unlike in the recursive setup, `core_1` and `core_2` do not share weights. The MeSH mechanism, with its memory buffer and routers, is then inserted at the boundaries between these conceptual blocks to manage information flow. Results are shown in Table 8. The `vanilla+mesh` model achieves a lower perplexity (7.26) than the standard `Vanilla` baseline (7.44), confirming that MeSH provides a direct performance uplift even without the constraint of parameter sharing. This finding provides a compelling explanation for the strong performance of our main `recursive+mesh` model: the

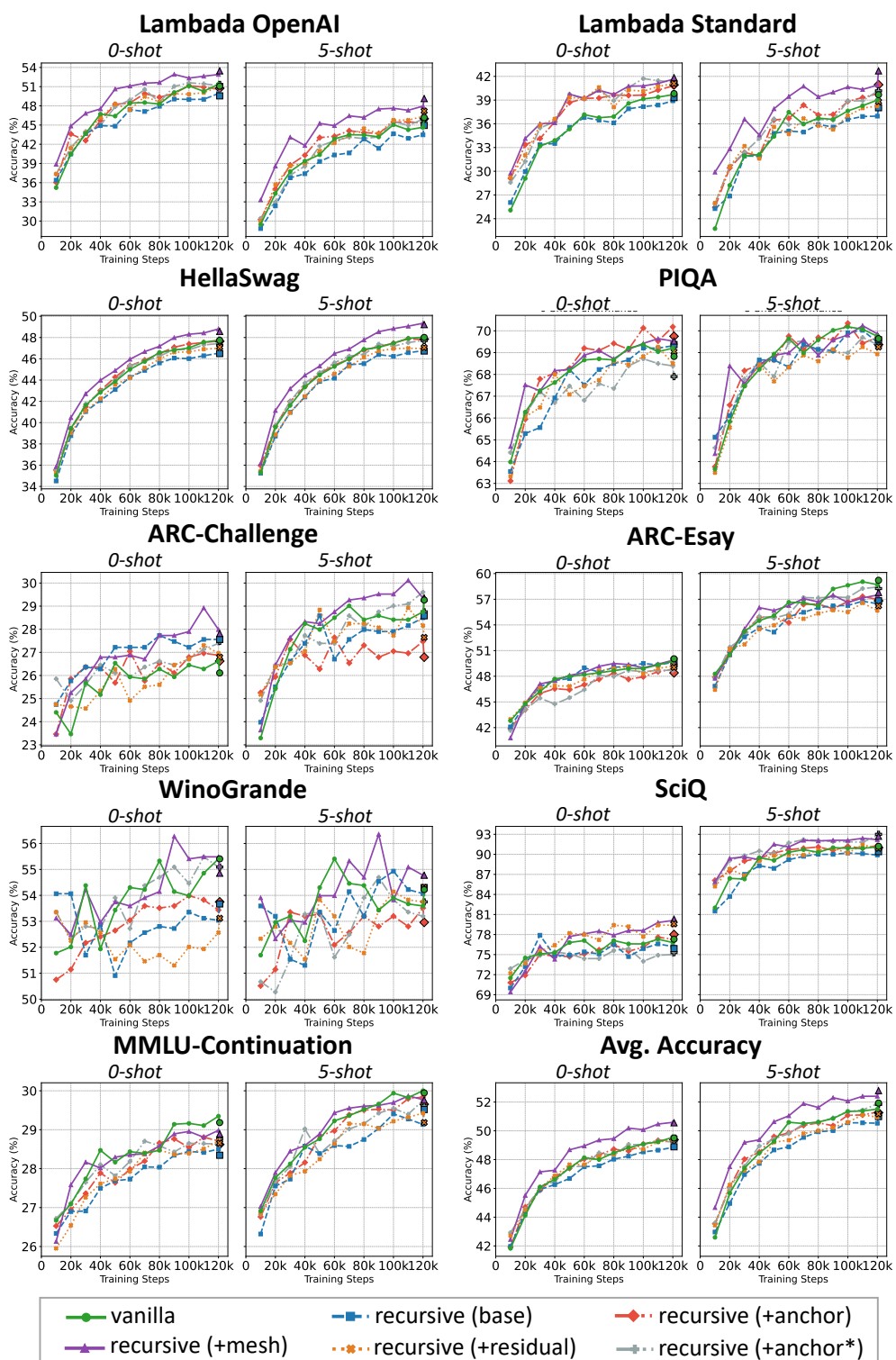

Figure 9: **Detailed Training Dynamics of 1.4B Recursive Variants on Downstream Tasks.** Each panel displays the 0-shot and 5-shot accuracy for one of the 9 individual downstream tasks or their overall average ("Avg. Accuracy"), evaluated at different checkpoints throughout the 120,818-step pre-training process.

architectural benefits of MeSH are potent enough to not only compensate for the performance loss typically incurred by weight sharing but to exceed the original baseline.

Table 8: Performance comparison of applying MeSH to recursive and non-recursive (Vanilla) backbones on the Pythia-1.4B scale. All metrics are evaluated on the Pile dataset.

| Variant | Config | Non-Emb Params (%) | Loss ↓ | PPL ↓ |
|---|---|---|---|---|
| vanilla | 24 layers | 100% | 2.0070 | 7.4406 |
| vanilla+mesh | 4+8+8+4 | 100% | **1.9818** | **7.2559** |
| recursive (base) | 4+8R2+4 | 66.7% | 2.0317 | 7.6267 |
| recursive+mesh | 4+8R2+4 | 66.7% | 1.9996 | 7.3865 |

While our MeSH framework was considered in recursive transformers, the result indicates that its core principle of explicit, routed state management has broader applicability. Exploring MeSH as a general architectural primitive for enhancing deep, non-recursive transformers is a promising direction for our future research.

### E.7 APPLYING MeSH TO MoE-BASED ARCHITECTURES

Table 9: **Performance of MeSH on a MoE-based backbone.** Results for a 2.6B-parameter OL-MoE model with 512M activated parameters. The recursive variants use a `4+8R2+4` configuration. PPL↓ is reported on Pile, Wikitext (Wiki), Lambda-OpenAI (LD-O) and Lambada-Standard (LD-S). Task Avg. acc↑ is reported for 0-shot and 5-shot settings.

| Structure | | | Perplexity↓ | | | | Task Avg. acc↑ (%) | |
|---|---|---|---|---|---|---|---|---|
| Scheme | Layers | Variant | Pile | Wiki | LD-O | LD-S | 0-shot | 5-shot |
| Vanilla | 24 | – | 7.31 | 14.93 | 11.29 | 22.29 | 49.83 | 51.87 |
| Recursive (-33.3%) | 4+8R2+4 | base | 7.60 | 15.97 | 11.99 | 22.52 | 48.96 | 50.83 |
| | | +mesh | 7.46 | 15.72 | 11.61 | 22.40 | 49.51 | 51.53 |

Mixture-of-Experts (MoE) models (Fedus et al., 2022; Dai et al., 2024), as a common technique in large language models (Liu et al., 2024; Llama Team, 2025; Yang et al., 2025), also leverage dynamic routing. However, the objective of this routing differs significantly: while MoE aims for computational sparsity by selecting a subset of experts within a feed-forward layer, MeSH is specifically designed to manage information flow across iterations in recursive architectures. This makes them orthogonal approaches that can be complementary in principle. To empirically validate that MeSH remains effective on a MoE-based recursive backbone, we adapted the open-source OLMoE architecture (Muennighoff et al., 2024). Our non-recursive vanilla baseline is an OLMoE model with 2.6B total and 512M activated parameters, configured with a hidden size of 1024 and 24 layers. For the recursive variants, we adapted this non-recursive backbone into a 4+8R2+4 configuration, aligning with our main experimental setup. The MoE-specific settings, such as using dropless token-choice routing to select 2 out of 16 experts (with an expert hidden dimension of 2048), were kept consistent. All models were pre-trained on the Pile dataset with a learning rate of 3e-4. As shown in Table 9, the results confirm that while naive recursion (base) leads to performance degradation, the +mesh variant successfully mitigates this drop. It recovers performance to a level comparable to the vanilla counterpart under matched compute, demonstrating MeSH's effectiveness on MoE-based backbones.

### E.8 SUPPLEMENTAL ANALYSIS: SHARED SUBSPACE COLLAPSE

To provide a more direct and intuitive visualization of the *loop representational collapse* identified in our main analysis (Figure 1c), we conduct a supplemental subspace analysis based on Singular Value Decomposition (SVD). Our central hypothesis is that the pathology we term **information overload** that leads to **representational conflict**, forcing the naive recursive model to make a dire trade-off. To ensure stability, the model sacrifices its role as an "Information Processor" and degenerates into a

simple "Information Preserver", forcing all iterative states to converge to a simple, low-dimensional "common ground" representation.

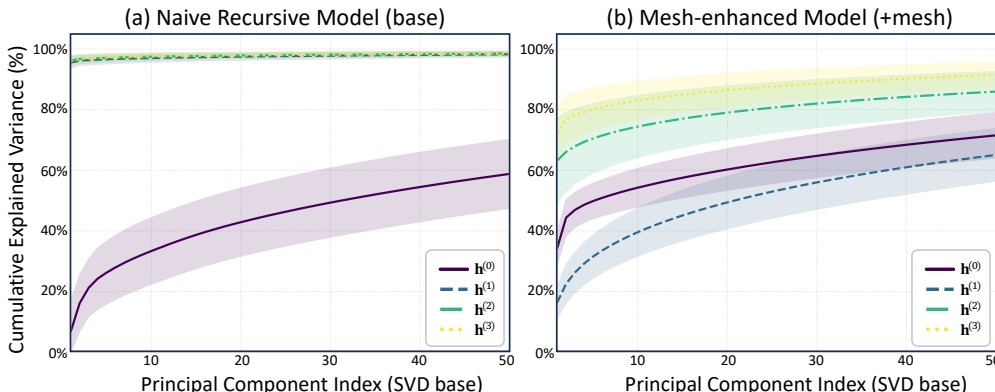

Figure 10:  **Subspace Analysis.** Visualization results of how information is structured within the recursive loop for **(a)** a naive recursive model and **(b)** our MeSH-enhanced model. Both models are based on the Pythia-410M backbone with a `3+6R3+3` recursive configuration. We first define a "Shared Subspace" by performing SVD on the concatenated hidden states of all loop iterations $(\mathbf{h}^{(0)}, \mathbf{h}^{(1)}, \mathbf{h}^{(2)}, \mathbf{h}^{(3)})$ and using the top singular directions as the shared basis. The plots then show the cumulative variance of each individual state when projected onto this shared subspace. In **(a)**, the states of the naive model dramatically collapse onto the shared subspace, where variance is almost entirely explained by the first few singular directions, indicating severe structural redundancy and a failure to generate new information. In **(b)**, the states of the MeSH model maintain distinct, high-dimensional structures, where variance is distributed across many components, demonstrating that each iteration preserves unique information and avoids collapse. Lines and shaded areas represent the mean and standard deviation across 500 samples.

As shown in Figure 10, we construct the shared subspace that captures the most dominant and consistent feature directions across all loop iterations. We then project each individual state back onto this shared basis and measure how much of its variance is explained. The results provide a stark contrast. In the naive recursive model (Figure 10a), the representations of all loop states ($\mathbf{h}^{(1)}$ through $\mathbf{h}^{(3)}$) collapse onto the shared subspace. Their variance curves are extremely steep and nearly identical, indicating that their entire structure is almost completely contained within the first few dimensions of the shared subspace. This provides compelling evidence of profound structural redundancy. The model is not generating new, diverse information at each step; instead, it is trapped in a low-dimensional attractor, merely preserving a static set of features. This is the hallmark of a system that has abandoned its "Processor" role. Conversely, the MeSH-enhanced model (Figure 10b) completely averts this pathology. Each state exhibits a distinct and gracefully rising curve, signifying that each state preserves a significant amount of unique, high-dimensional information not captured by the others. By offloading the duty of information preservation to its external memory, MeSH liberates the hidden states to engage in meaningful, high-dimensional computation at each step. The result provides visual evidence that MeSH resolves the representational bottleneck caused by the representational conflict, enabling progressive refinement of information across the recursive loop.

## F  CASE STUDY: UNPACKING THE INTERNAL DYNAMICS OF MESH

To provide a more granular, qualitative view of MeSH's internal mechanisms, we provide a case study on a single input sequence. Figure 11 unpacks the model's dynamics, showing how MeSH organizes information flow across recursive iterations. Panel (a) visualizes the router weights, revealing how the model learns to route information with both iteration-level and token-level specificity. The 'Write' router learns a clear policy: on average, it directs the output of each iteration to a distinct memory slot (e.g., Iteration 1 targets Slot 1, Iteration 2 targets Slot 4), where the memory organizes information from different computational stage. Furthermore, the stark difference between the average weights and those for the specific token highlights that routing is a highly dynamic, token-wise

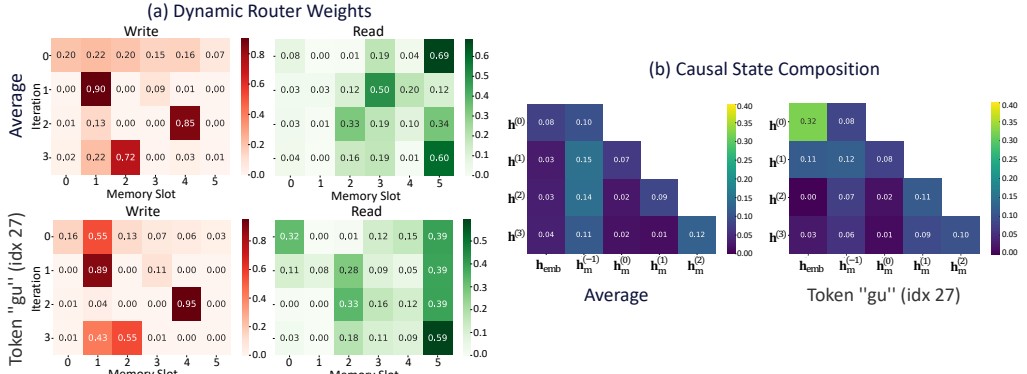

Figure 11: **Visualization of Internal Dynamics.** The analysis is performed on a MeSH-enhanced Pythia-410M model with a 3+6R3+3 configuration. Case study: "The failure of his handcrafted novel to become a bestseller was the ultimate wake-up call, forcing him to second-guess everything." The visualizations show: **(a)** Write (red) and Read (green) router weights, shown for the sequence average and for an intermediate token "gu" (idx 27). **(b)** State composition analysis. The coefficients shown quantify the causal contribution of each source state ($\mathbf{h}_{\mathrm{emb}}$, $\mathbf{h}_{\mathrm{m}}^{(i)}$) to the formation of a subsequent input state ($\mathbf{h}^{(t)}$). These are derived from the router weights and reflect the unrolled recurrence relation as Eq. 7. Here, $\mathbf{h}_{\mathrm{m}}^{(-1)}$ is the prelude output, and $\mathbf{h}_{\mathrm{m}}^{(t)}$ is the core output at loop iteration $t$.

decision. The state composition analysis in panel (b) shows how information is integrated across iterations. For instance, the state $\mathbf{h}^{(3)}$, which serves as the input to the final loop, is composed of information from the prelude output $\mathbf{h}_{\mathrm{m}}^{(-1)}$ as well as the core outputs from previous loops and $\mathbf{h}_{\mathrm{m}}^{(1)}$. This illustrates a sophisticated recombination strategy, where the model maintains a strong connection to the initial "anchor-like" state while flexibly incorporating information from intermediate steps, thereby effectively managing the information flow throughout the recursive process.

# G  LIMITATIONS AND FUTURE WORK

While this work establishes MeSH as a promising architectural principle for recursive transformers, we recognize several limitations that open up avenues for future research. First, our experiments have validated the effectiveness of MeSH on models up to the Pythia-6.9B scale, trained on the deduplicated Pile dataset. It remains unclear whether the parameter-efficiency gains persist at larger scales and under different training regimes. A natural and important direction for future work is to apply and evaluate the MeSH architecture on state-of-the-art foundation models at much larger scales. Second, although our ablation study reveals that MeSH can also benefit non-recursive transformers, a comprehensive investigation beyond recursive backbones falls outside the scope of this paper. Exploring MeSH as a general-purpose architectural primitive for improving information flow remains a promising direction for our future work.

# H  STATEMENT ON LARGE LANGUAGE MODELS USAGE

We used large language models only for language editing to improve the clarity and readability of the paper. The LLMs did not contribute to the research ideation, methodology and the experimental implementation. All substantive content is original and was verified by the authors.

