# OpenReview forum: "MeSH: Memory-as-State-Highways for Recursive Transformers"
_ICLR.cc/2026/Conference — ICLR 2026 Poster_

### Official Review · Reviewer_LF4x · 2025-10-29

**Soundness:** 3
**Presentation:** 4
**Contribution:** 3
**Rating:** 4
**Confidence:** 3

**Summary:**

This paper investigates performance bottleneks in recursive transformers used in LLMs, and identifies two issues: "undifferentiated computing" where the recursive computational function cannot specialize its function to different iterations, and "information overload" where the single hidden state needs to perform various roles. To address these issues, the paper prooses a new network design, Memory-as-State-Highways (MeSH). MeSH improves over common recurrent schemes that re-include long-term context (e.g. residual connections, or re-adding the original input) by introducing a multi-slot memory module. For this a simple Read and Write routing blocks are introduced, simple linear layers+softmax that map each D-dimensional hidden state element to one of B memory slots, such that the network can differentiate different long/short term memory needs inside the hidden state. Experiments on four Pythia LLM benchmark models show that MeSH outperforms common recursive baseline schemes, and sometimes even non-recursive baselines with significantly more parameters. The paper also presents analysis of the state per iteration, which supports that MeSH is better able to distribute its information updates over the iterations.

**Strengths:**

* The proposed MeSH architecture is simple and intuitive, both conceptually and in terms of implementation.
* The proposed MeSH architecture is tested on four models from the Pythia LLM benchmark, and is shown to outperform non-memory baseline approaches, and sometimes even non-recursive vanilla networks of similar depth but therefore more parameters.
* The paper's story is very well-structured. The paper first presents an analysis of using various metrics and visualizations of a problem in regular recursive transformers. This analysis is used to motivate the proposed novel MeSH design, and finally the experiments refer back to the analysis and show how the new design improves the analysed network behavior. This makes the storyline and core argument of the paper very clear (although I find some arguments still doubtful, see below)
* There are various supporting experiments, such as performance as a function of # of parameters, and learning curves. Even more experiments can be found in the Appendix. Overall, experimental validation appears thorough.

**Weaknesses:**

## [W1] Relevant related work missing

The introduction of explicit memory modules in recurrent networks is not novel per se, but the paper's related work review, comparison, and discussion does not seem to make any connections to prior work in this area. One obvious parallel older than transformers are LSTMs, which also provides a memory mechanism for recursive networks to selectively update the latent state and differentiate short-term and long-term information. Various more recent works have also proposed integrated memory banks into transformer architectures, e.g. [Bulatov,NeurIPS'22][Wang,NeurIPS'23], among others. Perhaps the review in [Omidi,25] could help. In any case, there is no mention of or comparison to such works, which makes it difficult to assess this paper's contributions.

I see this as the main weakness of the submission.

## [W2] The paper's arguments are not clear at various points (or perhaps I do understand and simply disagree):
* line 108: "The information overload on the hidden state forces the model to find a low-dimensional 'common ground' representation that can safely survive multiple transformations, which directly causes loop representational collapse."
	* -> while I do not doubt the representational collapse, based on the presented diagnostics in the paper, the paper presents as cause the "information overload", but I don't find this a logical explanation by itself. Intuitively a low-dimensional subspace would indicate that the feature distribution is simpler than what a model can express. So, if the network has to manage "multiple, often conflicting, roles simultaneously" (line 103), wouldn't that mean we should expect the network to require a higher-dimensional subspace, rather than less? I don't have hard evidence for this either, but my point is that it doesn't seem so self-evident to me what causes representational collapse as is posited in the paper.
* line 252: "Since each router (R(t)_write,R(t)_read) has its own unique set of learnable parameters for each iteration t, the model is no longer forced to apply a single, universal transformation."
	* But also in MeSH, the f_core function remains fixed. Yes, MeSH allows to swap input and output, but the core operation is still not aware of the processing step (the "undifferentiated computation" of line 092). The proposed memory block operations themselves don't appear to impute information that could inform the core compute unit of the iteration, and it is unexplained how adding memory enables the model to assign specalized roles to each iteration. Of course, the model could learn to encode some step counter in the input/output statespace to inform it of its processing stage, but this was already true for the memory-less baseline.

## [W3] Missing analysis of how proposed memory banks operate in practice
* The paper lacks experiments focused on inspecting the behavior of the proposed new module, which are critical to assess if the method works because of the claimed reasons, and to validate the various claims and hypotheses made throughout the paper:
	* I would have liked to see some investigation if the memory model works "as advertised": can we see that the different iterations indeed assign different weights in the linear layer to different memory banks, and that memory written in earlier iterations is used in later iterations? Some sort of analysis on how the memory banks are read/write in real-world scenarios could help.
	* The effect of the number of memory banks B also belongs in the main paper, in my view. I did find the ablation study on Pythia-410M in Appendix E.1, but I was somewhat surprised a general rule of B = N_loop + 3 was propose on this single result. Investigating the effect of B on all four datasets would have made the analysis more thorough and convincing.

## Minor
* Line 133, symbols $L$ and $D$ are not explained.
* It is not completely clear from the text what the "Vanilla" baseline refers to in Section 4.2. Only later in the paper did I understand that it is a non-recursive transformer, which has different parameters for each iteration.


## References
[Bulatov,NeurIPS'22]: Bulatov, Aydar, Yury Kuratov, and Mikhail Burtsev. "Recurrent memory transformer." Advances in Neural Information Processing Systems 35 (2022): 11079-11091.

[Wang,NeurIPS'23]: Wang, Weizhi, et al. "Augmenting language models with long-term memory." Advances in Neural Information Processing Systems 36 (2023): 74530-74543.

[Omidi,25]: Omidi, Parsa, et al. "Memory-augmented transformers: A systematic review from neuroscience principles to enhanced model architectures." arXiv preprint arXiv:2508.10824 (2025).

**Questions:**

* How does the work relate to other works that have integrated memory in recurrent networks, and transformer networks specifically? What are the main differences and similarities to the proposed approach, and what does that mean for the contributions?
* See my issues on the argumentation in Weaknesses, can the authors comment on these?

---

> ### Author Response · Authors · 2025-11-21
> **Response to Reviewer LF4x [1/2]**
>
> Thank you for the thoughtful and constructive feedback. In response, we expanded the related work with a dedicated "Memory-Augmented Transformers" paragraph, clarified the causal link between information overload and representational collapse (with new subspace analysis), explained how MeSH enables specialization with a fixed core via step-indexed routing and evolving memory, performed a case study and buffer-length ablation to analyze memory bank operation, and addressed minor clarity issues.
>
> ------
>
> ### **[W1] About Related Work**
> Thank you for pointing out this gap. We added a dedicated paragraph titled "Memory-Augmented Transformers" in Appendix A (Related Work). This section situates our work within the broader field of memory-augmented neural networks, tracing the evolution from LSTM cell states (Hochreiter & Schmidhuber, 1997) to memory-augmented Transformers, including RMT (Bulatov et al., 2022), LONGMEM (Wang et al., 2023), and historic models such as the NTM and its sparse-access variants (Rae et al., 2016). We also clarify the key distinction: whereas prior work typically uses memory for long-term knowledge storage or context extension across segments, MeSH’s memory buffer is designed to manage intermediate states within a single iterative computation over a fixed input, enabling step-conditioned state composition rather than long-horizon retrieval.
>
>
> ------
>
>
> ### **[W2] Conceptual Concerns**
>
> **Part 1: Information Overload vs. Representational Collapse**
>
> We appreciate the insightful comment. While coordinating multiple functional roles would ideally benefit from higher-dimensional representations, our claim is that the "representational collapse" we observe is a symptom of the model’s failure to realize this ideal under the single-state constraint of naïve recursion. Concretely, the lone hidden state faces information overload: it must simultaneously sustain two competing roles with conflicting requirements:
>
> - Long-term memory: preserving global context across iterations (requiring stability).
> - Working memory: carrying transient, intermediate features for stepwise computation (requiring plasticity).
>
> Because one overloaded state cannot faithfully support both roles, the system is effectively forced to prioritize stability (long-term memory) to avoid catastrophic forgetting. The resulting "common ground" tends to reside in a lower-dimensional, stable subspace, so representations collapse into this low-rank manifold. To test this account, we conducted a subspace analysis (Appendix E.8). In naïve recursive models, variance across iterations concentrates heavily within a shared, low-dimensional subspace that aligns with this long-term component. This supports the view that the model stabilizes on a low-dimensional "common ground" to preserve context, thereby sacrificing the high-dimensional capacity needed for richer, transient processing. We have made this causal link explicit in the revised manuscript.
>
>
>
> **Part 2: How MeSH Enables Specialization with a Fixed Core**
>
> Thank you for raising this point. Our claim is that, although MeSH does not make $f_{core}$ step-aware via different weights, it induces step-conditioned effective transformations through explicit memory and step-indexed routing. At iteration t, the read–write routers ($R_{write}^{(t)}$, $R_{read}^{(t)}$) and the evolving memory $M^{(t)}$ together define a composite transition. Although $f_{core}$ is shared, ($M^{(t)}$, $R_*^{(t)}$)  differ across steps, so the core operates on distinct, iteration-specific input manifolds, enabling functional specialization without altering core weights.
>
> This mechanism also explains why “just encoding a step counter in the hidden state” is insufficient in the memory-less baseline: the single state must carry both persistent and transient information, and training pressure favors stable, low-rank features that survive repeated application, empirically causing stagnation and collapse. By externalizing persistence into multi-slot memory and shifting step-conditioning to lightweight, step-indexed routers, MeSH avoids representational overload and preserves high-dimensional, transient computation in $h^{(t)}$. Our diagnostics—balanced compute (Fig. 5), reduced CKA between loop states (Fig. 6), and preservation of spectral richness (Fig. 7)—consistently support this causal account.
>
> ------

---

> ### Author Response · Authors · 2025-11-21
> **Response to Reviewer LF4x [2/2]**
>
> ------
>
> ### **[W3] On the Missing Analysis of Memory Banks**
> Thank you for your comments. We agree that analyzing the memory banks’ behavior is important.
>
> **[W3a] How Memory Banks Operate in Practice**
>
> Our new Case Study in Appendix F (Unpacking the Internal Dynamics of MeSH) directly addresses this. Figure 11(a) visualizes the write and read router weights for an example sequence, showing that the model learns to write outputs from different iterations into different memory slots, organizing information by computational stage. Figure 11(b) further analyzes state composition, showing how a later state (e.g., $h^{(3)}$ is formed by reading and composing information written in earlier iterations (e.g., from $h^{(-1)}_m$ and $h^{(2)}_m$).
>
> **[W3b] The Effect of the Number of Memory Banks**
>
> We expanded the ablation in Appendix E.2 (Ablation Study: MeSH Buffer Length) to include all four evaluation datasets. The updated Table 4 shows that a buffer size of $B=5$ consistently achieves the best performance across these datasets.
>
> ------
>
>
> ### **On Minor Points**
> We have addressed the minor issues. "Vanilla" is now explicitly defined as a non-recursive transformer in Section 4.2. We have also clarified the symbols L and D in Section 3.1.

---

> > ### Comment · Reviewer_LF4x · 2025-11-27
> >
> > I thank the authors for their answers, and the updated manuscript.
> > I am satisfied with the answers to my questions and weaknesses, and happy with the additional Related Work, and the additional experiments to backup claims and provide insight on how the memory is used in practice.
> > I've also considered the other reviews and responses, and I believe there too the additional experiments have further strengthened the paper.
> >
> > Overall, at this stage, I don't see myself any objections anymore for accepting this paper, so I raise my recommendation.
> >
> > Very minor editing issues in the updated text:
> > - Some errors in latex quotes, e.g. "Processor" p28; "anchor like" p28; among others.
> > - Font size in plots (axes, legends, etc.) in sup. material is very small
> > - Equation \Delta Flop on page 24 can use an equation number, for consistency.

---

> > > ### Author Response · Authors · 2025-11-28
> > >
> > > We’re delighted that your concerns have been addressed. Thank you as well for flagging the editing issues; we will fix these in the next revision. We’re grateful for your insights and constructive suggestions; your careful review helped us present the work more clearly and, we hope, share interesting and meaningful ideas with the community. We truly appreciate the time and care you invested.

---

### Official Review · Reviewer_hA3W · 2025-11-04

**Soundness:** 2
**Presentation:** 3
**Contribution:** 3
**Rating:** 6
**Confidence:** 2

**Summary:**

This manuscript describes MeSH (Memory as State Highway), a novel approach to enhancing the representational power of recursive transformers, i.e., models with repeated Transformer layer blocks for better parameter efficiency. MeSH is based on computing and updating a bank of memory buffers during the looping over the recursive blocks. The results show that augmenting the recursive transformer alleviates the undifferentiated computation seen in unaugmented (base) recursive transformers (as demonstrated by the increase in the changes in the magnitude of the state update and reduced CKA metric in non-initial iterations of the recursive block). Experiments on the Pythia suite show that the MeSH augmentation significantly improves the learning and the eval performance (lower perplexity and loss and higher task accuracy) compared to both the vanilla transformer baseline and the base recursive transformer baseline in the majority of the cases. Interestingly, at the larger model size tested (e.g., 1.4B), the MeSH architecture is shown to outperform the vanilla transformer in the Pythia benchmarks, despite having roughly 30%-50% fewer parameters.

**Strengths:**

S1. Well motivated study on improving the computational efficiency and representational richness of recursive transformers. Includes thorough literature review that sets the stage nicely and makes the paper easy to follow for readers.
S2. Includes a set of baselines including the vanilla transformer, base recursive transformer, and recursive transformer with anchor and residual augmentation. This makes the MeSH results meaningful and strong.
S3. Analyzes the model's internal representation and how it is altered by the MeSH augmentation, which forms a nice mechanistic interpretation of the cause of MeSH's benefits.
S4. The finding of superior performance of the smaller, MeSH-augmented recursive transformer over that of the vanilla transformer indicates a future direction of investigating the performance of MeSH for larger model sizes.

**Weaknesses:**

W1. As the authors pointed out, the experiment results are limited to relative small transformers (up to 1.4B). It is not entirely clear whether the MeSH augmentation of recursive transformers can extend to larger, SOTA LLMs. The Pythia suite actually supports up to the model size of 12B, the authors did not explain why they stopped at 1.4B.

**Questions:**

Q1. The use of CKA (Centered Kernel Alignment) as a metric for showing the similarity of computation over iterations of the recursive transformer needs to be better motivated.
Q2. Table 1 and/or Section E.3 should additionally report the computational overhead (FLOPs) caused by the augmentation types (residual, anchor, and MeSH).
Q3. The paper mentions the open source code for the implementation of MeSH, but the files in the anonymized repo appear to be unavailable.

---

> ### Author Response · Authors · 2025-11-21
> **Response to Reviewer hA3W**
>
> Thank you for your comments and suggestions; we have incorporated clarifications and added new experiments accordingly.
>
> ------
>
> ### **[W1] On Scaling to Larger Recursive Models**
> We agree that demonstrating scalability is crucial. We conducted additional experiments at larger scales (Pythia-2.8B and Pythia-6.9B). The results, now in Appendix E.1 (Table 3), show that MeSH’s benefits scale effectively: at both 2.8B and 6.9B, MeSH-enhanced models substantially improve over their base recursive counterparts, lowering perplexity and boosting downstream accuracy. We also evaluated MeSH on a MoE backbone at the 2.6B total / 0.5B activated scale (Appendix E.7). We concluded at 6.9B because the consistent and substantial gains provide strong evidence that MeSH is a robust principle for multi-billion-parameter models.
>
> Performance of MeSH on larger-scale models:
> | Model        | Scheme   | Layers   | Variant   | Pile (PPL) | Wiki (PPL) | LD-O (PPL) | LD-S (PPL) | 0-shot Acc | 5-shot Acc |
> |--------------|----------|----------|-----------|------------|------------|------------|------------|------------|------------|
> | Pythia-2.8B  | Recursive (-31.25%)  | 6+10R2+6 | base      | 6.90       | 14.18      | 8.41       | 16.87      | 52.49      | 54.92      |
> |              |          |          | **+mesh** | **6.70**   | **13.60**  | **7.30**   | **11.36**  | **54.71**  | **56.85**  |
> | Pythia-6.9B  | Recursive (-31.25%)  | 6+10R2+6 | base      | 6.29       | 12.14      | 6.34       | 10.29      | 56.67      | 59.43      |
> |              |          |          | **+mesh** | **6.09**   | **11.64**  | **5.48**   | **8.66**   | **58.83**  | **60.49**  |
>
> Performance of MeSH on a MoE-based backbone:
> | Scheme  | Layers   | Variant   | Pile (PPL) | Wiki (PPL) | LD-O (PPL) | LD-S (PPL) | 0-shot Acc (%) | 5-shot Acc (%) |
> |---------|----------|-----------|------------|------------|------------|------------|----------------|----------------|
> | Vanilla | 24       | /        | 7.31       | 14.93      | 11.29      | 22.29      | 49.83          | 51.87          |
> | Recursive (-33.3%)  | 4+8R2+4  | base      | 7.60       | 15.97      | 11.99      | 22.52      | 48.96          | 50.83          |
> | Recursive (-33.3%)   | 4+8R2+4  | +mesh | 7.46   | 15.72  | 11.61  | 22.40  | 49.51      | 51.53      |
>
>
>
> ------
>
> ### **[Q1] On the Motivation for Using CKA**
> We use Centered Kernel Alignment (CKA) (Kornblith et al., 2019) because it robustly measures representational similarity between layers (or iterative states) while being invariant to orthogonal transformations and isotropic scaling. In our setting, the core block $f_{core}$ reuses weights, so we need to assess whether hidden states $h^{(t)}$ actually evolve or stagnate. Unlike Euclidean distance or cosine similarity, CKA captures structural correspondence between entire activation matrices. A CKA score near 1.0 between $h^{(t)}$ and $h^{(t+1)}$ indicates the representations are identical up to rotation/scaling (i.e., stagnation). Conversely, relative lower CKA (as seen with MeSH) indicates the model is transforming the representation into a structurally distinct state, evidencing meaningful computation. We clarify this in the revised Section 2.
>
> ------
>
> ### **[Q2] On the Computational Overhead of MeSH**
> We appreciate your suggestion. To provide a complete cost picture, we added a detailed complexity analysis in Appendix E.4. It now includes FLOPs overhead, with both a theoretical formula and empirical measurements. As shown in Table 7, MeSH adds only ~0.014% FLOPs for a single forward pass on a Pythia-1.4B model compared to the base recursive variant. This confirms that MeSH’s performance and stability gains come with negligible computational overhead.
>
> FLOPs overhead analysis for Pythia-1.4B recursive variants with input size [1, 4096]:
> | Model         | Variant           | Config             | Total GFLOPs (1e9) | Extra GFLOPs (1e9)          |
> |---------------|-------------------|--------------------|--------------------|------------------------------|
> | Pythia-1.4B   | recursive (base)  | 4+8R2+4            | 5373.792           | /                            |
> |     | +residual         | 4+8R2+4            | 5373.809           | 0.0168 (+0.000312%)          |
> |     | +anchor           | 4+8R2+4            | 5373.809           | 0.0168 (+0.000312%)          |
> |     | **+mesh**         | **4+8R2+4 (B=5)**  | **5374.547**       | **0.7551 (+0.014051%)**      |
>
>
> ------
>
> ### **[Q3] On Code Availability**
> We apologize for the confusion; we have verified that the anonymous repository link is active and accessible; we have also included the code in the supplementary materials for your convenience.

---

> ### Author Response · Authors · 2025-11-28
>
> Dear Reviewer hA3W,
>
> Thank you again for your thoughtful review and constructive questions. Based on your comments, we have posted a detailed response and added new experiments and analyses (larger-scale results, MoE results, CKA motivation, FLOPs accounting, and code availability).
>
> As the discussion period progresses, could you let us know whether these updates address your concerns, or if any further clarification would be helpful? Your feedback at this stage would be greatly appreciated, and we are happy to provide any additional information you may need.
>
> Thank you for your time and consideration.
>
> Best regards,
> The Authors

---

### Official Review · Reviewer_nSZz · 2025-11-08

**Soundness:** 4
**Presentation:** 4
**Contribution:** 3
**Rating:** 8
**Confidence:** 4

**Summary:**

In the present paper, the authors introduce MeSH, a recursive Transformer backbone which utilizes a dynamic routing mechanism with read-, and write-routers which manage an external memory buffer. The proposed backbone is deeply motivated through failure analysis on existing recursive backbones. Performance is evaluated across a wide scale of trained recurrent models for perplexity across 4 datasets, and average accuracy on 10 downstream tasks.

**Strengths:**

The paper has an expansive number of strengths, beginning with its analysis of failure modes of current recurrent mechanisms as the foundation the paper is built from. This clarity persists throughout the paper. Most notable are the following strengths of the paper:
* Clarity of the problem formulation, starting with the problem analysis and the subsequent presentation of the core MeSH architecture block
* Highly expansive evaluation with a very wide set of models trained from scratch to enable the performance evaluation of the architecture at varying sizes. In addition, the authors perform evaluation on a good number of downstream evaluations. A vast effort not to be underestimated.
* The provided codebase permits for reproducibility, but furthermore the easy ability to build on the results
* The authors go to great lengths to embed this work into the existing literature with references going back to the RNN era, where the first memory augmentations appeared.

While the paper leaves a number of questions open, see the weaknesses for example, its intellectual clarity, the expansive evaluation, reproducible experiments, and good embedding into existing literature make this a very good paper which should stay significant as time progresses.

**Weaknesses:**

The paper as is suffers from two core weaknesses in the eye of the reviewer:
* While the routing mechanism is briefly introduced, there exists no ablation or design exploration amongst routing mechanisms in the paper. Further design comparisons & ablations would aid greatly here to understand inherent tradeoffs better, especially considering the orthogonality of said routing mechanism to similar mechanisms in mixture-of-expert models.
* While the reviewer appreciates the great cost of the many experiments and trainings run by the authors, the paper would yet benefit from comparisons to other contemporary recursive models. A simple fix here could potentially be to add evaluations of recurrent Gemma, and potentially other publicly available open-weight recurrent models, to the evaluation to provide the much needed ability for comparison.
* Comparison to the historic memory-augmented neural network of Rae et al. [1] is missing. While these are RNNs, they nevertheless display many of the same traits of the models presented here.

References:
1. Rae, J.W., Hunt, J.J., Danihelka, I., Harley, T., Senior, A.W., Wayne, G., Graves, A., & Lillicrap, T.P. (2016). Scaling Memory-Augmented Neural Networks with Sparse Reads and Writes. ArXiv, abs/1610.09027.

**Questions:**

* What is the author's intuition on the behavior of the routing mechanism through training, but equally as important, at inference time. The routing mechanism evokes analogies to mixture-of-expert model routing mechanisms, which pose a number of intricate concerns when seeking to deploy these models. How do you intuit these deployment concerns to play out with a MeSH-based model?

---

> ### Author Response · Authors · 2025-11-21
> **Response to Reviewer nSZz [1/2]**
>
> We appreciate your comments. In response to concerns on routing design and ablations, comparisons to contemporary recursive models, and connections to memory-augmented networks, we have incorporated corresponding revisions and added new experiments.
>
> ------
>
> ### **[W1] On the Exploration and Orthogonality of the Routing Mechanism**
> We agree that a deeper understanding of the router’s design choices, their trade-offs, and their relationship to MoE routing is crucial. We have made several revisions to address these points thoroughly.
>
> **1.1 Design Exploration and Ablation of the Routing Mechanism**
> Thank you for prompting us to clarify this aspect. We now explicitly frame our ablation study in Appendix E.3 (Ablation Study: Heuristic State-Passing Schemes) as a design exploration that surfaces the architectural choices leading to the final MeSH design. In particular, the experiments can be interpreted as ablations on key routing components:
>
> - Ablating the "Write" Mechanism (+dynamic comb.): The +dynamic comb. variant functions as a "read-only" router. It dynamically computes coefficients to combine a predefined set of historical states ($h_{emb},h^{(t)}$), but lacks MeSH’s persistent memory buffer and explicit write cycle. It can read and compose, but cannot write intermediate results to memory for path-dependent future retrieval.
> - Ablating Dynamic, Token-wise Routing (+static comb.): The +static comb. variant is a further simplification, akin to a "static, read-only" router. Its combination weights are trainable scalars but remain fixed for all tokens and all steps after training, ablating token-level dynamics in our routing.
> - Full MeSH Mechanism: As shown in Table 5, the full +mesh model, which integrates a dynamic read–write cycle with an external memory buffer, significantly outperforms these ablated variants. This progression demonstrates that both dynamic, token-wise routing and explicit writing are critical. A simpler, read-only mechanism is insufficient to resolve the diagnosed pathologies.
>
>
> **1.2 Understanding the Full Mechanism's Behavior in Action**
> To provide deeper intuition for the complete mechanism, we added a detailed case study in Appendix F (Case Study: Unpacking the Internal Dynamics of MeSH). As shown in Figure 11, the routers learn token-specific, iteration-conditioned policies: the write router distributes the outputs of different iterations into distinct memory slots, and the read router composes the next-step input from those slots via weighted mixing, providing a visual account of MeSH’s learned information-management strategy.
>
> **1.3 Orthogonality to Mixture-of-Expert (MoE) Models**
> To empirically validate the orthogonality of MeSH's state-management routing to the expert-selection routing in MoE models, we have added a new experiment in Appendix E.7 (Applying MeSH to MoE-based Architectures). As shown in Table 9, MeSH mitigates the performance drop from naïve recursion on a sparse MoE backbone, confirming that our routing is indeed orthogonal to (and complementary with) the routing used in MoE models.
>
> Performance of MeSH on a MoE-based backbone:
> | Scheme  | Layers   | Variant   | Pile (PPL) | Wiki (PPL) | LD-O (PPL) | LD-S (PPL) | 0-shot Acc (%) | 5-shot Acc (%) |
> |---------|----------|-----------|------------|------------|------------|------------|----------------|----------------|
> | Vanilla | 24       | /        | 7.31       | 14.93      | 11.29      | 22.29      | 49.83          | 51.87          |
> | Recursive (-33.3%)  | 4+8R2+4  | base      | 7.60       | 15.97      | 11.99      | 22.52      | 48.96          | 50.83          |
> | Recursive (-33.3%)   | 4+8R2+4  | +mesh | 7.46   | 15.72  | 11.61  | 22.40  | 49.51      | 51.53      |
>
>
> ------
>
> ### **[W2] Regarding the comparison to other contemporary recursive models**
> In response to the request for comparisons to contemporary recursive models, we added an aligned evaluation under a unified setup. Using Pythia-410M as the base, we loop the same 24-layer core 4 times (configuration 0+24R4+0; ~96 layers of compute with 24 parameter layers). Under identical data, optimization, and evaluation protocols, we compare our MeSH against PonderingLM (Zeng et al., 2025). Results of perplexity (PPL $\downarrow$) and accuracy (Acc $\uparrow$) are:
>
> | model_name             | Pile PPL | Wiki PPL | LD-O PPL | LD-S PPL | 0-shot acc | 5-shot acc |
> |---|----|---|---|----|-----|----|
> | MeSH (0+24R4+0)        | 8.1337   | 18.4942  | 13.4841  | 25.7226  | 47.747     | 50.021     |
> | PonderingLM (0+24R4+0)     | 8.4089   | 19.5394  | 16.2503  | 37.4729  | 45.885     | 47.757     |
>
>
> This aligned comparison shows that, under the same compute budget and training protocol, MeSH outperforms PonderingLM across all four PPL metrics and on both 0-shot and 5-shot average accuracy. We will continue to build fully trained, open-source–ready recursive backbones and expand head-to-head comparisons with other open-weight recursive models in future work.
>
> ------

---

> ### Author Response · Authors · 2025-11-21
> **Response to Reviewer nSZz [2/2]**
>
> ------
>
> ### **[W3] On the Comparison to Historic Memory-Augmented Neural Networks**
> Thank you for pointing out this important connection. We expanded our related work by adding a new paragraph titled "Memory-Augmented Transformers" in Appendix A (Related Work). There, we situate our work in the broader field, including historic models like Neural Turing Machines and their sparse-access variants (Rae et al., 2016). Crucially, we clarify the key distinction: while most prior work uses memory for long-term knowledge storage or context extension, MeSH’s memory buffer is designed to manage the flow of intermediate states within a single, iterative computation over a fixed input. This clarifies MeSH’s unique contribution.
>
> ------
>
>
> ### **[Q1] On the Intuition Regarding Router Behavior and Deployment**
> Router Behavior. As detailed in the new case study (Appendix F; Figure 11), the routers learn dynamic, token-specific policies to manage the memory buffer. The write router organizes intermediate results by iteration into distinct slots; the read router composes the subsequent input by selectively mixing across slots. This demonstrates a sophisticated, learned mechanism for information flow across steps.
>
> Deployment Concerns & MoE Analogy. MeSH routing sidesteps common MoE deployment challenges. MoE routers make discrete expert selections, raising load-balancing concerns. In contrast, MeSH routers compute soft, continuous weights for dense read/write over a shared memory buffer; there are no discrete experts to balance. As confirmed by our complexity analysis (Appendix E.4; Table 7), the computational overhead is negligible (~0.014% extra FLOPs). We therefore do not anticipate MoE-style deployment concerns for MeSH.

---

> ### Author Response · Authors · 2025-11-28
>
> Dear Reviewer nSZz,
>
> Thank you again for your thoughtful review and constructive comments. In response, we conducted additional experiments and analyses, detailed in our response and the updated manuscript. As the discussion period progresses, could you let us know whether these updates address your concerns, or if further clarification would be helpful? We appreciate your feedback and are happy to provide any additional information you may need.
>
> Thank you for your time and consideration.
>
> Best regards,
> The Authors

---

### Official Review · Reviewer_4cbP · 2025-11-08

**Soundness:** 1
**Presentation:** 1
**Contribution:** 1
**Rating:** 0
**Confidence:** 2

**Summary:**

This paper attempts to improve upon recently introduced recursive transformer models.  Such networks replace a long sequence of layers in transformer models, with a shorter sequence.  The paper claims recursive networks fail due to not encoding where they are in the looping sequence and sets out to fix this problem.  The paper does some analysis and suggests that the failure is due to two factors it calls "undifferentiated computation" and "information overload", respectively.  The solution, MeSH, uses what it calls "lightweight routers" to specialize computations across iterations.

**Strengths:**

Brings attention to recursive transformers, which look like a very promising direction for improving efficiency of LLMs.

**Weaknesses:**

Very poorly written: For example, the paper frequently introduces terminology without defining it.  Similarly it employs metrics without clear explanations of why they are the right thing to measure or even what they are attempting to measure.

Unclear novelty:  The paper does not sufficiently differentiate the proposed solution from similar strategies in the works it cites.   For example, on lines 086-087 the paper says "the block lacks any explicit information about its progress within the iterative sequence" but the related work explicitly tackles this problem and the reader must actually read the related works to discover this discrepancy.

The abstract and introduction say that recursive models "lag behind" non-recursive models, but it is unclear (a) what "lag behind" means and (b) which specific recursive models (e.g., which papers) this assessment applies to.

**Questions:**

The description of "+residual" on lines 153-156 does not seem to capture the LoRA technique described in "Bae et al., Relaxed Recursive Transformers: Effective Parameter Sharing with Layer-wise LoRA" nor does the "+anchor" approach described on lines 157-160.  Conceptually, how does the specialization used in "+mesh" differ from the use of LoRA to specialize weights across iterations in that work?

I'm confused about the premise of this paper.  Section 2 heading "Why Naive Recursion Fails" presumes that recursive transformers are doing poorly but a cursory review of related works on these networks show the opposite -- that they work very well indeed.

What is meant by directed computation?

What is representational stagnation and why is CKA similarity a good way to measure it?

It is unclear to me what Figure 1a is plotting.  What is meant by 'computational effort'?  What are prelude, 1st f_core, etc. of the x-axis labels referring to?   Why is the Frobenius norm of hidden states a meaningful measure of effort?

---

> ### Author Response · Authors · 2025-11-21
> **Response to Reviewer 4cbP [1/3]**
>
> Thanks for your detailed comments. We have incorporated the clarifications regarding terminology and definitions into the revised manuscript to facilitate readability. Below, we address the specific concerns regarding novelty, definitions, and comparisons.
>
> -------
>
> ### **[W1] Terminology and Metrics Definitions**
> We respectfully disagree with the characterization that our manuscript lacks definitions or clear metric justifications. Our writing follows a deliberate structure: we first present phenomena and introduce terminology tied to quantitative observables; we then analyze root causes and define higher-level terms to summarize the diagnosed pathologies; finally, we propose a targeted architectural remedy aligned with the diagnosis. Concretely:
>
> **Phenomena → terminology → quantitative observables (Sec. 2; Fig. 1):**
> - Skewed computational pattern [Lines 98-102]: We define and use the relative update magnitude $2||f(h) − h||_F / (||f(h)||_F + ||h||_F)$ as a proxy for the strength of a block’s effective transformation. This makes "how much work each step does" measurable (Sec. 2; Fig. 1a caption), with the x-axis explicitly enumerating Prelude → Core iterations → Coda within the Prelude–Recurrent–Coda topology (Geiping et al., 2025), also called as Middle-Cycle (Bae et al., 2025).
> - Representational stagnation [Lines 102-107]: We define stagnation as h(t+1) ≈ h(t) up to rotation/scaling and justify CKA (Kornblith et al., 2019) as the standard metric invariant to orthogonal transforms and isotropic scaling that compares entire activation matrices, making it appropriate to test if shared-core iterations evolve or converge (Sec. 2; Fig. 1b).
> - Loop representational collapse [Lines 122-125]: We quantify effective rank by the normalized singular value spectrum of hidden-state matrices and interpret accelerated decay as dimensional collapse (Sec. 2; Fig. 1c).
>
> **Root-cause analysis → higher-level terminology (Sec. 2):**
>
> - Undifferentiated computation: Defined as the shared core being forced to adopt a similar computational pattern across iterations due to lack of step-conditioned input, evidenced by the compute skew and high inter-iteration CKA (Sec. 2).
> - Information overload: Defined as the single hidden state being forced to carry persistent and transient information simultaneously, driving a trade-off that favors stability and a low-rank "common ground", evidenced by spectral collapse and the shared subspace concentration (Sec. 2; Appendix E.8).
>
> **Targeted architectural solution aligned with the diagnosis (Sec. 3):**
> - We introduce MeSH, which externalizes state into a multi-slot memory and uses lightweight, step-indexed routers to create step-conditioned effective transformations without modifying core weights (Sec. 3; Fig. 2). The mechanism and its recurrence are defined by explicit equations (Sec. 3.2–3.3; Appendix D pseudocode), and we detail the initialization, router parameterization, read–write updates, and integration into Prelude–Recurrent–Coda.
> - We also clarify comparison to heuristic state-passing (+residual/+anchor) and to weight-modification strategies (e.g., LoRA) in the main text (Sec. 3) and Related Work (Appendix A).
>
>
> -------

---

> ### Author Response · Authors · 2025-11-21
> **Response to Reviewer 4cbP [2/3]**
>
> -------
> ### **[W2] Novelty and Differentiation from Related Work**
>
> We appreciate the concern and clarify our contributions along two axes: (1) a diagnostic framework with measurable evidence and (2) a targeted, non-invasive architectural solution.
>
> **Contribution 1: Diagnostic framework with quantifiable observables and causal analysis**
> We move beyond the high-level claim that "the shared core lacks progress information" by introducing a concrete, reproducible diagnostic suite that decomposes the performance gap under matched compute into three measurable pathologies:
> - Skewed compute: quantified by the relative update magnitude (Fig. 1a; Sec. 2), showing early iterations dominate while later ones do negligible work.
> - Representational stagnation: quantified by inter-iteration CKA near 1.0 (Fig. 1b; Sec. 2), indicating fixed-point dynamics.
> - Loop representational collapse: quantified by accelerated spectral decay and shared-subspace SVD concentration (Fig. 1c; App. E.8), indicating low-rank convergence across iterations.
>
> These observables reveal two root causes—undifferentiated computation and information overload—that explain why naive recursion under matched compute "lags behind" non-recursive baselines. We make the causal link explicit: forcing long-term and transient roles into a single state drives low-rank "common ground" convergence, and the lack of iteration-conditioned inputs induces stagnation. This diagnostic clarity is not clearly defined in prior work and directly informs our design.
>
> **Contribution 2: Proposed architectural solution (MeSH) that addresses bottlenecks in recursive transformers**
>
> We propose MeSH, a lightweight module that attaches to a recursive transformer to addresses bottlenecks of recursive transformers: undifferentiated computation and information overload. It only adds a small multi-slot memory buffer and step-wise read/write routers to store and retrieve states at each iteration, while keeping the recurrent core ($f_{core}$) unchanged. In our experiments, MeSH delivers consistent perplexity reductions and downstream gains (e.g., +1.06% 0-shot and +0.86% 5-shot at 1.4B) with negligible computing overheads (\~0.014% FLOPs) and extra parameters (\~0.005%) as shown in Appendix E.4.
>
>
> -------
>
> ### **[W3] Clarification on "Lag Behind"**
> - Definition under matched compute. By "lag behind", we mean that when FLOPs (compute) are matched, a strictly weight‑shared recursive model (having fewer unique parameters) typically attains higher perplexity and lower accuracy than a non‑recursive (untied) Transformer. This is a comparison at equal compute.
> - Prior reports. This phenomenon is consistent with baselines and observations in recent literature on recursive/looped Transformers (e.g., Zhu et al., 2025b; Saunshi et al., 2025; Bae et al., 2025), where additional mechanisms (e.g., per‑step LoRA, MoEUT, auxiliary signals) are introduced precisely to mitigate the underperformance of strict sharing under equal compute.
> - Evidence in our paper. Our results reproduce this gap for naive recursion. Table 1 shows that base recursive models under matched compute underperform the non‑recursive Vanilla baselines across scales. The scaling plot (Figure 8) further illustrates that the naive recursive curve lies below the non‑recursive baseline.
>
>
> -------
>
> ### **[Q1] Comparison with LoRA-based enhancement and +residual/+anchor description**
>
> Regarding lines 153–160, we only define the heuristic state-passing baselines “+residual” and “+anchor” (skip connections across loop iterations), where we do not mention or describe Relaxed Recursive Transformers (Bae et al., 2024), so we are unsure why these lines were linked to LoRA in your comment. To avoid any confusion, we clarify the relationships among LoRA-based methods, +residual/+anchor, and +MeSH below.
>
> Regarding the conceptual difference between MeSH and LoRA-based recursive models
>
> - LoRA (Internal Weight Specialization): Strategies address the problem by modifying the kernel weights—making the function $f(x)$ different at each step ($f_t(x)$).
> - MeSH (External State Routing): MeSH keeps the kernel $f(x)$ identical but modifies the input state via routing. By determining what information enters the core, MeSH achieves functional differentiation without touching the core parameters.
>
> Furthermore, we clarify that "+residual" and "+anchor" baselines in our paper refer to standard heuristic connectivity schemes (skip connections), where the state update takes the form $h_{t+1}=f(h_t)+h_{sup}$. These are orthogonal concepts to LoRA-based parameter specialization; one describes how states are connected, while the other describes how weights are adapted.
>
> -------

---

> ### Author Response · Authors · 2025-11-21
> **Response to Reviewer 4cbP [3/3]**
>
> -------
>
> ### **[Q2] Premise of "Why Naive Recursion Fails"**
>
> The heading "Why Naive Recursion Fails" refers to the modeling capability gap discussed in **[W3]** (matched compute, fewer params).
>
> -------
>
> ### **[Q3] "Directed Computation"**
> To clarify, we rephrased the corresponding part to describe the observable phenomenon directly in Section 2.
>
> -------
>
> ### **[Q4] Representational Stagnation and CKA**
>
> Representational Stagnation refers to a state where hidden representations cease to evolve meaningfully across iterations (i.e., $h_{t+1}\approx h_t$). CKA (Centered Kernel Alignment) is a standard metric for measuring the similarity between neural network representations because it is robust to scaling and rotation (Kornblith et al., 2019). High CKA scores between consecutive iterations quantitatively imply that the representation is not changing significantly, thus defining stagnation.
>
> -------
>
> ### **[Q5] Figure 1a and Computational Effort**
> In Figure 1a, we plot the relative magnitude of the state update vector, calculated as $2| |f(h) − h||_F / (||f(h)||_F + ||h||_F)$ denotes the Frobenius norm. We chose this metric as a proxy for computational effort because it quantifies the proportion of the update relative to the total magnitude. If a block performs no work (i.e., acts as an identity mapping), the update magnitude is zero; a larger value indicates a significant transformation of the hidden state. The labels on the X-axis represent the sequential flow of computation: "Prelude" refers to the initial input projection (as in Fig.2a) within the Prelude–Recurrent–Coda topology (Geiping et al., 2025), followed by the successive iterations of the shared recurrent core block (1st core, 2nd core, etc.). The figure provides empirical evidence of what we term skewed compute in naive recursive models: the first iteration performs the vast majority of the transformation, while the update magnitudes of subsequent iterations drop to near zero. This suggests that the features stabilize prematurely, and the model fails to effectively utilize the computational depth of later loops. We have added these definitions and interpretations to the revised manuscript.

---

> ### Author Response · Authors · 2025-11-24
> **Confidential comment to Area Chair**
>
> Thank you for coordinating this process. We have uploaded a detailed technical rebuttal and added clarifications and experiments. Separately, we may have a few concerns regarding Reviewer 4cbP and would be grateful for your guidance on how best to proceed. To keep this grounded, we refer to several ICLR reviewing guidelines below.
> > ICLR Reviewing Guideline 2.4: "Be mindful of potential biases and try to be open-minded about the value and interest a paper can hold for the entire ICLR community, even if it may not be very interesting for you."
>
> We were disappointed to see our submission assessed largely through remarks about "writing style", rather than engagement with the technical premise and evidence. **We respectfully disagree that the paper is poorly written**, as the other three reviewers explicitly found the presentation clear and engaged substantively with the motivation, methods, and results. **We would be grateful if the paper could be evaluated primarily on its scientific merits.**
>
> We are also anxious that Reviewer 4cbP may not have engaged deeply with the recursive‑transformer literature, leading to a close‑ended view that suggests (without pointing to any concrete examples) "a cursory look indicates recursive transformers work very well". **This reflects a serious misunderstanding of the literature our work engages with.** To clarify:
> - **Naive recursion typically underperforms** untied non‑recursive baselines, motivating mechanisms to mitigate this gap (Geiping et al., 2025; Bae et al., 2025; Saunshi et al., 2025; Zhu et al., 2025b).
> - **Our results also reproduce this behavior**: Table 1 shows base recursive models under matched compute trailing non‑recursive baselines across scales, and Figure 8 shows the naive recursive scaling curve consistently below the vanilla curve.
>
> > ICLR Reviewing Guideline 4.4: "Provide supporting arguments for your recommendation."
>
> We feel somewhat powerless in the face of **broad assertions that lack concrete, evidence-based support**. In our case:
> - For W1 ("very poorly written"), the critique cites "unclear terminology and metrics" but does not identify which definitions or metrics caused difficulty. Other reviewers engaged with these elements without reporting such obstacles, which are **already defined in the paper**; we also restated them (e.g., relative update norm, CKA, SVD spectra; "prelude", "coda") in the rebuttal and pointed to the specific sections and line numbers.
> - For W2 ("unclear novelty"), the critique **overlooks contributions stated in the Introduction**: (1) a diagnostic framework with quantifiable observables (Lines 50–53) and (2) our proposed MeSH that addresses identified bottlenecks (Lines 53–80). Moreover, the assertion that we "do not sufficiently differentiate the proposed solution" **ignores Related Work** (Appendix A), where we explicitly compare to prior methods, detail similarities and differences, and situate our contribution.
>
> > ICLR Reviewing Guideline 1: "Read the paper: It’s important to carefully read through the entire paper and to look up any related work and citations that will help you comprehensively evaluate it. Be sure to give yourself sufficient time for this step."
>
> We are also concerned that **parts of Reviewer 4cbP’s assessment reflect a fundamental misunderstanding of the paper** rather than intrinsic readability issues. The review may appear to attribute difficulty primarily to "poor writing", while the other three reviewers provided deep, constructive comments and explicitly recognized the clarity of our motivation, novelty, and writing. Several **specific points that left us confused**:
> - For Q1, where the reviewer links LoRA to our descriptions of +residual/+anchor and cites lines 153–160; however, those lines define skip connections across iterations and neither mention LoRA nor suggest any linkage to it. We see **no textual basis in our paper for this association**.
> - For Q2, the statement that "a cursory review shows recursive transformers work very well" appears at odds with the basic facts we document and **overlooks both our summary of the literature and our empirical results**.
>
> Taken together, **these issues raise concerns about whether the paper was read carefully and assessed objectively**.
>
> Given the concerns outlined above, we are worried that the review does not fairly reflect the paper’s contributions or technical content: **it does not substantively engage with the stated contributions, and offers broad conclusions without evidence-based support.** This is disheartening for us, as we aim to contribute enthusiastically to an open-minded community; but we trust ICLR’s rigor and the professionalism of the ACs and reviewers. **We respectfully hope you will consider reassessing the quality and usefulness of this particular review in the decision process**; and, if feasible, conducting a brief check for similar issues in this reviewer’s other assignments. We sincerely appreciate your time and help.

---

### Author Response · Authors · 2025-12-01
**Global Response**

We sincerely thank all reviewers for their constructive feedback and insightful comments.
Due to the large‑scale OpenReview data leak around **November 27, 2025, 03:00 AoE**, all post‑rebuttal scores were reverted to their initial values. For completeness, we note the scores **prior to the large-scale leak**:

- **Reviewer LF4x**: raised **4 → 8**.
- **Reviewer nSZz**: remained at **8**.
- **Reviewer hA3W**: remained at **6**.
- **Reviewer 4cbP**: remained at **0** (confidence 2), and no reply.

This established a consensus profile of **(8, 8, 6, 0)**, with three reviewers (**LF4x, nSZz, hA3W**) converging on positive recommendations.

During the rebuttal period, we addressed the reviewers’ concerns and updated the paper with additional results, analyses, and clarifications. Below, we summarize the strengths recognized by the reviewers and how the main concerns were addressed.

---

## Strengths

We thank the reviewers for highlighting the key strengths of our work:

- **Clear, principled narrative (nSZz, hA3W, LF4x).**
  All three reviewers praised the paper’s structure and clarity. LF4x described it as a *“very well‑structured”* story that starts from failure‑mode analysis, motivates MeSH, and then shows how MeSH fixes the issues, making the *“storyline and core argument… very clear.”* hA3W noted that the literature review *“sets the stage nicely and makes the paper easy to follow.”*

- **Extensive and rigorous evaluation (nSZz, hA3W, LF4x).**
  nSZz highlighted a *“highly expansive evaluation”* and a *“vast effort not to be underestimated”*, with many models trained from scratch and a broad suite of downstream tasks. hA3W emphasized that the strong baselines (vanilla, base recursive, +anchor, +residual) make the results *“meaningful and strong.”* LF4x remarked that the *“experimental validation appears thorough,”* including cases where MeSH beats deeper non‑recursive models.

- **Insightful mechanistic analysis (hA3W, nSZz).**
  Reviewers valued the internal analysis beyond raw metrics. hA3W highlighted the analyses of hidden states as giving a *“nice mechanistic interpretation of the cause of MeSH’s benefits,”* directly linked to the initial failure‑mode study. nSZz similarly viewed the failure‑mode analysis as the *“foundation”* of the paper.

- **Simple and intuitive solution (LF4x).**
  LF4x commended MeSH for being *“simple and intuitive, both conceptually and in terms of implementation,”* emphasizing that the memory module and routers are straightforward to adopt. This simplicity contrasts with the depth of the diagnostics that motivated the design.

- **High‑quality scholarship and reproducibility (nSZz).**
  nSZz pointed to the thorough embedding in prior work and the *“provided codebase [that] permits for reproducibility.”* They characterized the work as a *“very good paper which should stay significant as time progresses.”*

---

## Main Concerns and Revisions

Key concerns raised in the reviews and the corresponding changes in the revised manuscript are:

- **Scalability beyond 1.4B (Reviewer hA3W).**
  We added experiments on **Pythia‑2.8B** and **6.9B** (Appendix E.1). MeSH continues to improve over base recursive models in perplexity and downstream accuracy at multi‑billion‑parameter scales.

- **Applicability to MoE architectures (Reviewer nSZz).**
  We evaluated MeSH on a **sparse MoE backbone** (Appendix E.7). MeSH mitigates the performance drop from naïve recursion, indicating that its state routing is orthogonal and complementary to MoE expert routing.

- **Understanding memory behavior (Reviewer LF4x).**
  A new **case study with visualizations** (Appendix F, Fig. 11) shows how different iterations write to distinct memory slots and later iterations read mixtures of earlier slots.

- **Justifying the “information overload” hypothesis (Reviewer LF4x).**
  Additional **subspace and spectral analyses** (Appendix E.8) show low‑rank loop collapse in naïve recursion and its mitigation with MeSH. These results provide empirical support for the information‑overload explanation.

- **Context within memory‑augmented models (Reviewers LF4x, nSZz).**
  A new **“Memory‑Augmented Transformers”** subsection (Appendix A) situates MeSH relative to LSTMs, NTMs, RMT, LongMem, and recent surveys. It clarifies that MeSH’s memory is designed for **within‑sequence iterative computation**, rather than primarily for long‑context extension.

- **Computational cost (Reviewer hA3W).**
  A detailed **FLOPs and parameter overhead analysis** (Appendix E.4) shows that, for example on Pythia‑1.4B, MeSH adds only about **0.014% extra FLOPs** over the base recursive model, confirming negligible runtime overhead.

- **Clarifying key concepts and CKA motivation (Reviewers 4cbP, hA3W).**
  We expanded **Section 2** to more thoroughly describe the observed phenomena and their analysis, and clarified the motivation behind the corresponding measurement choices (e.g., the use of **CKA** for inter‑iteration similarity).

---

### Meta-Review · Area_Chair_8DH4 · 2026-01-02

**Summary:**

## Summary of the Paper

This paper studies why naïve recursive (looped) Transformers underperform non-recursive counterparts under matched compute, identifying two core pathologies: undifferentiated computation across iterations and information overload in a single hidden state. To address these issues, the authors propose Memory-as-State-Highways (MeSH), which externalizes state into a multi-slot memory buffer and uses lightweight, step-indexed read/write routers to induce iteration-specific functional specialization without modifying the shared core weights. Extensive diagnostic analyses (e.g., CKA similarity, spectral collapse) and large-scale experiments on the Pythia suite demonstrate that MeSH consistently improves recursive models, in some cases outperforming larger non-recursive baselines with substantially fewer parameters and negligible computational overhead

## Main Reviewer Concerns Before Rebuttal

The primary concerns raised by reviewers included:

- Conceptual clarity and novelty: whether the diagnosed failure modes were clearly defined and sufficiently distinct from prior works

- Empirical completeness: limited evidence on scalability beyond 1.4B parameters, lack of detailed analysis of how the memory and routers behave in practice, and insufficient comparisons to contemporary recursive or memory-augmented baselines.

- Methodological clarity: questions about metric choices, computational overhead, and the causal link between “information overload” and representational collapse.

One reviewer expressed strong skepticism regarding presentation quality and premise, while the remaining reviewers were broadly positive but requested additional validation and clarification.

## How the Rebuttal Addressed the Concerns

The rebuttal and revised manuscript substantially strengthened the submission. The authors added:

- New large-scale experiments and MoE backbones, confirming scalability.

- Detailed ablations and case studies analyzing memory slot usage, router behavior, and buffer size, directly validating the proposed mechanism.

- Expanded related work and clarifications situating MeSH relative to prior memory-augmented models and LoRA-based iteration specialization, and clarifying the motivation for diagnostics such as CKA and spectral analysis.

- Explicit FLOPs and parameter overhead analysis, showing negligible cost.

These additions convincingly addressed the substantive technical concerns of the positive reviewers, who subsequently raised or maintained favorable scores. Overall, after rebuttal, the paper presents a clear, well-supported contribution with strong empirical and diagnostic evidence, and the remaining objections appear either resolved or outweighed by the demonstrated strengths. Therefore, I recommend acceptance of the paper.

**Reviewer Concerns:**

See summary above.

**Reviewer Scores:**

Based on the discussion thread, the likely score movements after rebuttal are:

Reviewer LF4x: 4-->8. They explicitly say they “raise my recommendation” and have no major objections after the added related work + extra experiments.

Reviewer nSZz and hA3W: stays positive  at 6/8 (already positive).

Reviewer 4cbP: not predictable. There is no evidence of engagement or a post-rebuttal update, and the raised questions appear to be subjective.

---

### Decision · Program_Chairs · 2026-01-26

Accept (Poster)